# Inducing novel endosymbioses by implanting bacteria in fungi

Gabriel H. Giger[1], Chantal Ernst[1], Ingrid Richter[2], Thomas Gassler[1], Christopher M. Field[1], Anna Sintsova[1], Patrick Kiefer[1], Christoph G. Gäbelein[1,6], Orane Guillaume–Gentil[1], Kirstin Scherlach[2], Miriam Bortfeld-Miller[1], Tomaso Zambelli[3], Shinichi Sunagawa[1], Markus Künzler[1], Christian Hertweck[2,4,5] & Julia A. Vorholt[1]✉

Endosymbioses have profoundly impacted the evolution of life and continue to shape the ecology of a wide range of species. They give rise to new combinations of biochemical capabilities that promote innovation and diversification[1,2]. Despite the many examples of known endosymbioses across the tree of life, their de novo emergence is rare and challenging to uncover in retrospect[3–5]. Here we implant bacteria into the filamentous fungus *Rhizopus microsporus* to follow the fate of artificially induced endosymbioses. Whereas *Escherichia coli* implanted into the cytosol induced septum formation, effectively halting endosymbiogenesis, *Mycetohabitans rhizoxinica* was transmitted vertically to the progeny at a low frequency. Continuous positive selection on endosymbiosis mitigated initial fitness constraints by several orders of magnitude upon adaptive evolution. Phenotypic changes were underscored by the accumulation of mutations in the host as the system stabilized. The bacterium produced rhizoxin congeners in its new host, demonstrating the transfer of a metabolic function through induced endosymbiosis. Single-cell implantation thus provides a powerful experimental approach to study critical events at the onset of endosymbiogenesis and opens opportunities for synthetic approaches towards designing endosymbioses with desired traits.

Intracellular endosymbioses are extraordinarily intimate interactions between organisms. They join two complex metabolic networks in a compartmentalized manner and are subjected to natural selection as a unit. This metabolic integration predisposes endosymbioses to enable major transitions in evolution[1,2]. Accommodating an endosymbiont can benefit the host by acquiring chemical defence systems and unlocking essential nutrients or new energy sources[2,6–8]. However, many obstacles limit de novo endosymbiogenesis[3–5]. Besides the first hurdle of host cell entry, a prospective endosymbiont must overcome challenges associated with immune responses, metabolism and growth synchronization[3,9,10]. Even if the combined metabolisms theoretically sustain growth in silico[11], unstable outcomes are the prevailing norm, failing to stabilize vertical transmission[11,12].

Studying established natural partnerships has provided insights into the intricate interactions of extant endosymbioses. These include mitochondria and chloroplasts as relics of ancient bacterial endosymbionts, as well as insect endosymbionts that have long been vertically transmitted and have undergone genome reduction[13]. However, shifting balances of control between the partners, phases of stabilization and destabilization and equivocal lines between mutualism and antagonism blur evolutionary trajectories[13–16]. Consequently, the earliest steps in endosymbiogenesis remain difficult to uncover[15]. Synthetic approaches can provide well-defined starting points to follow stable and unstable outcomes. Interventional studies have focused mainly on insects, in which transinfection has revealed important aspects of endosymbiogenesis, such as control of bacterial replication[17,18] and metabolic cooperation[19,20]. However, attempts in other phyla, such as installing engineered *E. coli* or cyanobacteria in yeast[21,22], have not resulted in stable endosymbiosis under strict selection.

Here we set out to generate a novel endosymbiotic partnership in a non-host filamentous fungus. The model system consisted of the early divergent fungus *Rhizopus microsporus* and the cytosolic bacterial endosymbiont *Mycetohabitans rhizoxinica*[23,24]. Certain host strains of *R. microsporus* contain *Mycetohabitans* endosymbionts[25]. In these, vertical inheritance is strict, as the fungus cannot sporulate without the endosymbiont, which reliably colonizes the spores[26,27]. This probably drives co-diversification[25]. The association is presumed mutually beneficial[28], with bacterially produced rhizoxin congeners[29,30] providing the fungus fitness advantages by protecting against amoeba and nematodes[31], and aiding in nutrient acquisition by causing rice seedling blight[23]. Both partners can be cultured axenically, and host strains cured from the endosymbiont are readily reinfected by *M. rhizoxinica*[26,32]. By contrast, non-host strains resist natural colonization and do not require endosymbionts for sporangiospore formation[33,34]. For this study, *R. microsporus* strain EH (endosymbiont-harbouring) and strain NH (non-harbouring) were used.

[1]Institute of Microbiology, Department of Biology, ETH Zurich, Zurich, Switzerland. [2]Department of Biomolecular Chemistry, Leibniz Institute for Natural Product Research and Infection Biology, HKI, Jena, Germany. [3]Laboratory of Biosensors and Bioelectronics, Institute for Biomedical Engineering, ETH Zurich, Zurich, Switzerland. [4]Institute of Microbiology, Faculty of Biological Sciences, Friedrich Schiller University Jena, Jena, Germany. [5]Cluster of Excellence Balance of the Microverse, Friedrich Schiller University Jena, Jena, Germany. [6]Present address: Whitehead Institute, Cambridge, MA, USA. ✉e-mail: jvorholt@ethz.ch

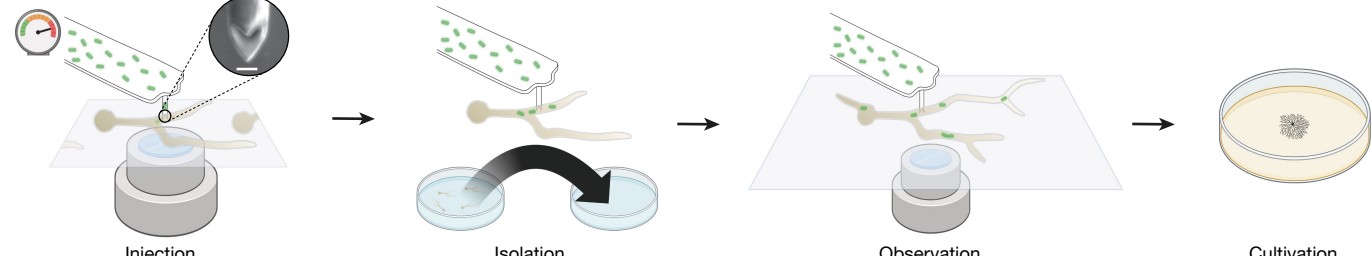

**Fig. 1 | Workflow for bacteria injection into fungal germlings.** FluidFM injection takes place on a glass dish surface placed above an inverted confocal light microscope. The probe apex is sharpened to a double point with an aperture of 500 to 1,000 nm. Inset: focused ion beam image of probe apex; scale bar, 500 nm. The turgor of the hypha is overcome by applying up to 6.5 bar overpressure. After the injection, the still-attached germling can be isolated into an empty dish, in which the recovery and growth of the microorganisms can be observed. The injected germling can subsequently be detached from the probe and transferred to a Petri dish for further cultivation until sporulation. Created with BioRender.com.

To study initial endosymbiogenesis events, we used single-cell approaches to observe cellular responses before environmental selection could act. Fluidic force microscopy (FluidFM)[35] was recently adapted to inject bacteria into mammalian cells, bypassing cellular entry steps and enabling evaluation of engineered pairs to test intracellular growth[36,37]. Applying this technique to fungi is challenging owing to the complexity of fungal mycelia, the rigid cell wall and high turgor pressure[38,39]. In this work, we report a procedure to implant bacteria into *R. microsporus* that enabled real-time tracking using confocal microscopy and characterizing early adaptations in the endosymbiosis under stabilizing selection pressure.

## Bacteria delivery into fungal hyphae

We have previously shown that with FluidFM, bacteria can be injected into mammalian cells[36,37] and small molecules can be injected into fungi[39]. However, the physical placement of bacteria into a fungal hypha had not been achieved with this technique and is experimentally more challenging. We used cylindrical tips with aperture sizes of 500–1,000 nm to allow the passage of bacteria while minimizing wound size. To improve our ability to puncture fungal cells, we sharpened the apex of the cylindrical FluidFM probes into a double point using focused ion beam milling (Fig. 1) and increased the applied force. In addition, the cell wall of the germlings was softened using an enzyme mixture in osmoprotective solution. Despite this treatment, pressures up to 6.5 bar were necessary for injection. With this approach, we were able to implant about 1–30 bacteria per injection event into an *R. microsporus* germling. The workflow also allowed isolation of injected germlings, subsequent imaging of fluorescently tagged bacteria inside the host cell, and culturing of the isolated germlings on agar plates to grow them until sporulation (Fig. 1).

Injection of *M. rhizoxinica* labelled with green fluorescent protein (GFP) into its natural host *R. microsporus* strain EH confirmed that both partners survived the injection procedure. Colonization was validated by microscopic observation (Supplementary Video 1). On the basis of the velocity, bidirectionality, the tracks followed and the motion in relation to the general cytoplasmic contents, the bacteria are probably moving both by active transport along microtubules and by passive transport through cytoplasmic bulk flow (Supplementary Videos 2 and 3). Microtubules have been shown to be involved in the transportation of *Wolbachia*-containing vesicles in *Drosophila*[40], but not in the transport of directly cytosolic endosymbionts or in fungi. We speculate that *M. rhizoxinica* may be able to attach to the microtubule-dependent motors dyneins and kinesins, through fungal cargo adaptor proteins or hitchhiking adaptor proteins[41]. The inhibition of dynein by transient exposure to ciliobrevin D, a reversible and cell-permeable small molecule, led to the transient cessation of bacterial transport within the germling (Supplementary Video 4).

To follow the expected vertical transmission of the endosymbiont in this reconstituted natural endosymbiosis, we used fluorescence-activated cell sorting (FACS) and microscopy to accurately classify large numbers of spores for their colonization status with fluorescently labelled bacteria. This analysis confirmed the stable inheritance of the implanted bacteria in spores.

## Injection of *E. coli* into *Rhizopus*

After reconstituting the natural endosymbiosis as proof of concept, we investigated the fate of the a priori non-endosymbiotic but intracellularly viable bacterium, *E. coli*, inside *R. microsporus*. *E. coli* has been shown to multiply rapidly in the cytosol of mammalian cells, resulting in host cell killing[36]. We injected GFP-labelled *E. coli* into *R. microsporus* strain EH and strain NH (Fig. 2a and Supplementary Videos 5 and 6). After injection, both partners remained viable, but *E. coli* intracellular replication and localization dynamics differed substantially from that of *M. rhizoxinica* in its natural host. *E. coli* proliferated more rapidly within the hyphae, dividing bacteria generally stayed together, and the resulting bacterial clumps dispersed only slowly, following the cytoplasmic bulk flow within hyphae (Supplementary Videos 5 and 6). Combined, these dynamics led to the emergence of local hotspots with high bacterial density. The fungi formed septa around these densely populated areas, resulting in compartments with and without *E. coli*. Bacteria-free compartments continued to grow normally, whereas compartments with bacteria became entirely filled, and occasionally burst open (Supplementary Videos 5 and 6). Defence by septa formation has previously been observed with *R. microsporus* strain EH responding to *M. rhizoxinica* mutants lacking the effector *Mycetohabitans* transcription activator-like effector 1 (MTAL1), which presumably helps the bacterium elude fungal defences[42]. Although some *E. coli* cells dispersed within individual hyphae over larger distances, leading to the formation of septa at sites remote from the original physical delivery (Fig. 2a and Supplementary Video 5), no *E. coli* could be detected in the spores collected from injected fungi, as evaluated by FACS (detection limit 1/1,000,000 positive spores, $n = 3$; Fig. 2c). These experiments showed that the fungus recognized unadapted bacterial intruders and triggered a defence reaction to physically contain the bacteria in separated hyphal compartments, allowing growth to proceed normally in uninfected hyphae.

## Vertical transmission of *M. rhizoxinica*

We next sought to generate a novel endosymbiosis by implanting GFP-labelled *M. rhizoxinica* into *R. microsporus* strain NH, which naturally does not harbour endosymbionts and readily forms spores without bacterial presence. After injection, the bacteria divided and exhibited transport dynamics similar to those observed in its host strain

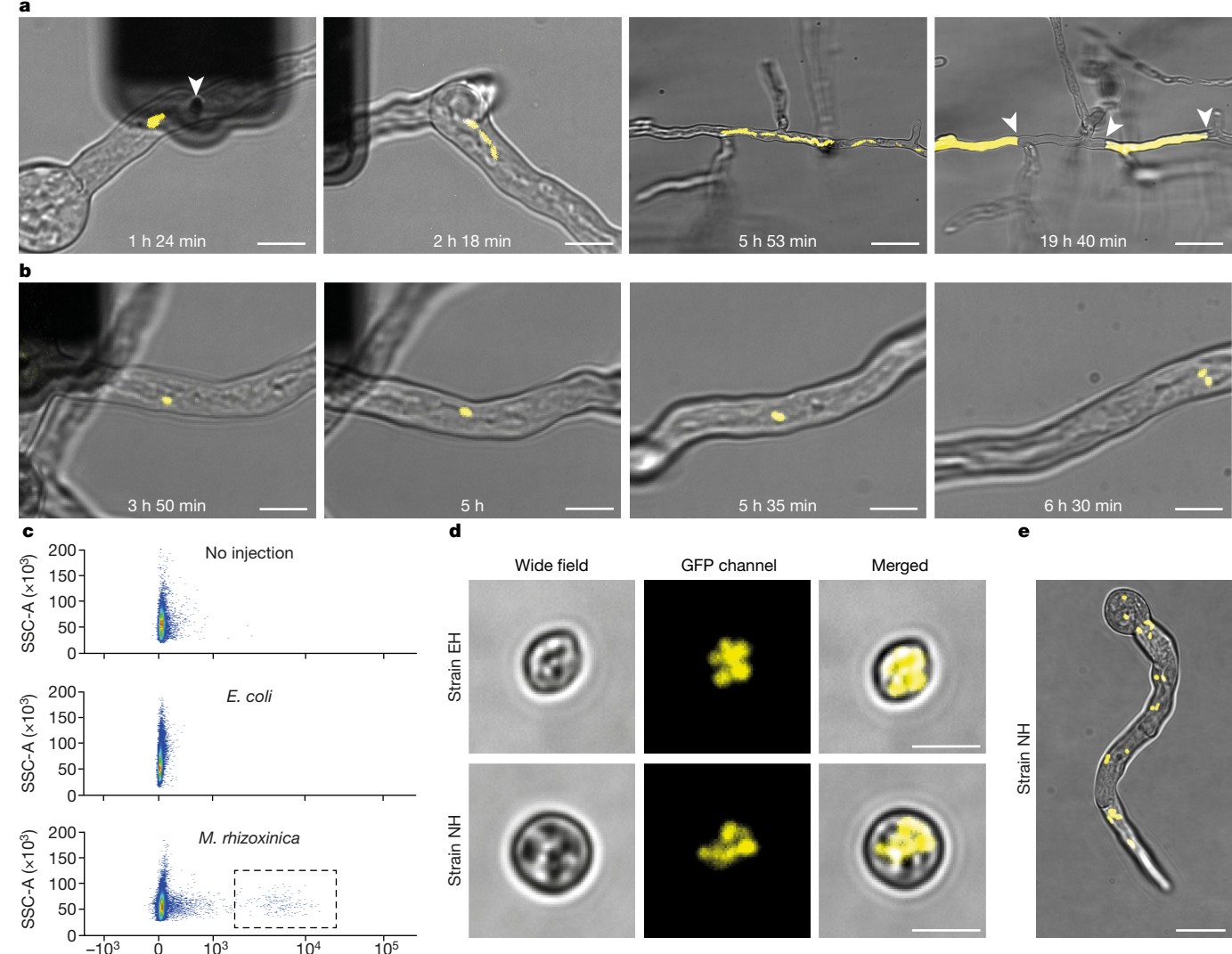

**Fig. 2 | Injection of *M. rhizoxinica*, but not *E. coli*, leads to vertical transmission of endosymbionts. a**, Images show the growth of the fungus and *E. coli* cells (yellow) at different time points. In the left image, the dark shape is the FluidFM probe, and the white arrowhead marks the apex of the probe inserted in the germling. In the right image, the white arrowheads indicate formed septa preventing bacterial spread. Images with differing magnification have different illumination settings and contrast adjustments. *E. coli* injections were carried out three times each in *R. microsporus* strain EH and strain NH with similar results (total *n* = 6). Shown here is strain EH. **b**, *M. rhizoxinica* replicates in *R. microsporus* strain NH (see also Supplementary Video 4). The experiment was carried out four times with similar results. **c**, Flow cytometry plots for spores collected from *R. microsporus* strain NH after no bacteria injection (top), *E. coli* injection (middle) and *M. rhizoxinica* injection (bottom). After injection with *M. rhizoxinica*, a high-GFP-signal fraction is observed (dashed grey rectangle), stemming from labelled bacteria. The experiment was carried out three times with similar results. Samples were run on different days on the same machine. **d**, Images showing positively FACS-sorted spores with intracellular bacteria. Top: host *R. microsporus* strain EH populated with the natural endosymbiont. Bottom: injected *R. microsporus* strain NH populated with *M. rhizoxinica*. The experiment was carried out four times with similar results. **e**, Image showing a germling from the positively sorted spore fraction of *R. microsporus* strain NH injected with *M. rhizoxinica*. The bacteria are distributed within the whole germling and are present in higher density than directly after an injection. The experiment was carried out four times with similar results. For **a**,**b**,**d**,**e**, images show a single-*z*-layer wide-field image overlaid with the two-dimensional projection of the GFP-signal *z*-stack in yellow. Scale bars, 10 μm (**a**, two left images, **b**), 40 μm (**a**, two right images), 5 μm (**d**) and 20 μm (**e**).

EH (Fig. 2b and Supplementary Videos 7 and 8). FACS revealed that a subset of the spores produced by the injected and cultivated germlings were colonized by bacteria (Fig. 2c). We confirmed microscopically that the positively sorted spores (≤4%; Extended Data Table 1) indeed harboured bacteria (Fig. 2d). When exposed to rich medium, some of the positive spores germinated successfully and retained high numbers of replicating bacteria that spread through the hyphae (Fig. 2e). These findings showed that *M. rhizoxinica* can be vertically transmitted in *R. microsporus* strain NH, and that implantation by FluidFM combined with selection by FACS can expand the host range of an endosymbiotic bacterium to a new host.

## Adaptive evolution of endosymbiosis

Building on the observed vertical transmission of *M. rhizoxinica* implanted into a novel host, *R. microsporus* strain NH, we next conducted an adaptive laboratory evolution experiment. To exert a stringent selection pressure, spores containing bacteria were sorted by FACS and propagated through successive rounds of growth and selection (Fig. 3a). To start, spores originating from a single injected germling were sorted. The positive spores were split into ten lines and subsequently propagated separately for seven rounds (see Fig. 3a and Methods for details). Three high-performing lines, and the pooled

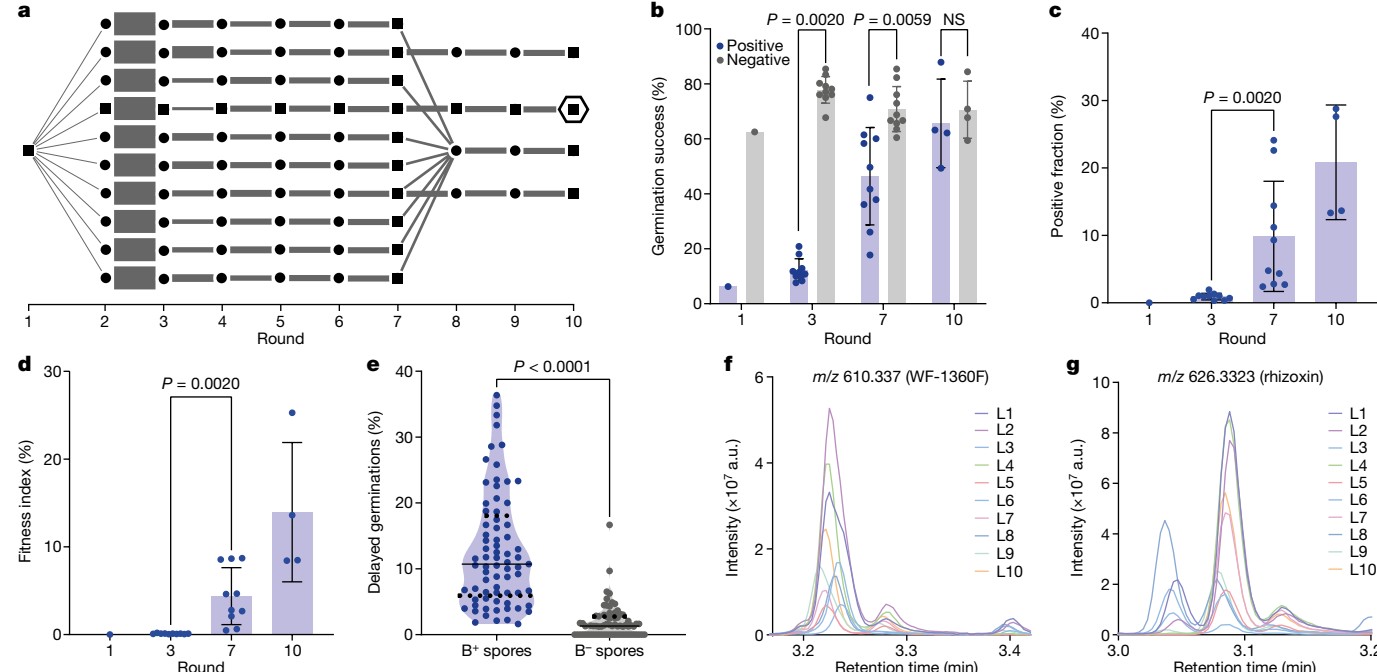

**Fig. 3 | Adaptive laboratory evolution experiment leads to higher fitness of the induced endosymbiosis. a**, Scheme of population sizes and line management throughout the experiment. Black dots indicate spore collection events with measurement of positive fraction and germination success, and sorting of positive spores. Black squares mark collection events in which, additionally, genomic DNA was isolated from the population and sequenced. Grey bars illustrate the number of positive spores plated. Round 1 spore collection is *R. microsporus* strain NH germling injected with *M. rhizoxinica*. The hexagon marks round 10 of line 4. **b**, The germination success of bacteria-positive spores (blue) and bacteria-negative spores (grey) at different rounds of propagation. NS, not significant. **c**, The positive fraction as measured by FACS increases over time. **d**, The fitness index, as calculated by multiplying the positive fraction from **c** with the germination success from **b**. **e**, Bacteria-positive (B⁺) spores exhibit an increased level of delayed germinations compared

to bacteria-negative (B⁻) spores throughout the directed evolution experiment. Solid lines indicate mean; dashed lines indicate quartiles. $n = 73$ plates. $P < 0.000000000001$. In **b**–**d**, for round 1, $n = 1$, round 3 and 7, $n = 10$, and round 10, $n = 4$ biological samples from one experiment. In **b**–**d**, data are presented as mean values ± s.d. In **b**–**e**, two-sided Wilcoxon matched-pairs signed rank test. **f**, The bacterially produced rhizoxin precursor WF-1360F can be detected in the induced endosymbiosis. **g**, The compound rhizoxin can be detected in the induced endosymbiosis. In **f**,**g**, plate extracts of all ten lines (L1–L10) from round 7 of the adaptive laboratory evolution experiment were analysed by liquid chromatography with tandem mass spectrometry. The molecules are detected in all ten lines. Shown are the sections of the liquid chromatography retention times of the highest signal intensity for *m/z* 610.337 (WF-1360F; **f**) and *m/z* 626.3323 (rhizoxin; **g**). a.u., arbitrary units.

remaining seven lines, were propagated for three more rounds. After each round of sporulation, we measured the percentage of the spores that contained bacteria (positive fraction), as well as the germination success of these spores. At the beginning of the evolution experiment, the germination success of positive spores was severely impaired. Whereas spores without bacteria had a germination success of 63%, the germination success of spores containing bacteria was reduced tenfold and was only 6.3% (Fig. 3b), suggesting that the presence of bacteria imposes a substantial cost on host fitness. The germination success of the positive spores had increased to 75% in the tenth round and was no longer distinguishable from that of spores without bacteria (Fig. 3b and Extended Data Fig. 1). Likewise, the positive fraction had increased from 0.01% to up to 24.1% at the end of the experiment (Fig. 3c). Both the germination success and the positive fraction directly affected the fitness of the endosymbiosis in an evolutionary sense (that is, the total amount of viable offspring with endosymbiont). We therefore defined the fitness index of the endosymbiosis outcome as the product of the two values. By round 7, the fitness index had increased from the initial 0.0006% to up to 8.7% with an average of 4.4% (Fig. 3d). The average fitness index in round 10 reached 14.0%, with the highest value noted for line 4 at 25.3%. An additional cost to the fungus observed over the course of the evolution was a delayed germination for spores with bacteria compared to that for spores without bacteria (Fig. 3e). Taken together, the findings of the adaptive laboratory evolution experiment showed that the

artificially induced endosymbiosis can be stably propagated in the new host upon positive selection. Over time the fitness index of the endosymbiosis increased, indicating adaptation towards improved transmission.

## Induced rhizoxin production in *Rhizopus*

We tested whether the incorporation of *M. rhizoxinica* as a new endosymbiont within the non-host strain NH led to the transfer of the biosynthesis of natural products, analogous to the natural host–endosymbiont system. In the natural system, *M. rhizoxinica* produces macrocyclic polyketides such as WF-1360F, which can be modified by *R. microsporus* to form rhizoxin and other congeners[29], and has been attributed to fitness advantages[23,29–31]. Rhizoxins are toxic to eukaryotic cells because they bind β-tubulin and interfere with microtubule formation[43]. The resistance of the fungus to rhizoxins is due to a single amino-acid substitution in β-tubulin that is present in most Mucorales, including both strains of *R. microsporus* used in this study[28]. We tested for the production of rhizoxin in the induced endosymbiosis with liquid chromatography with tandem mass spectrometry. All extracts from plates of the ten evolution lines in round 7 contained rhizoxin congeners. Specifically, WF-1360F and rhizoxin were both found in all lines (Fig. 3f,g). The detection of rhizoxin validates the transfer of a metabolic trait to strain NH, which does not naturally generate the natural product.

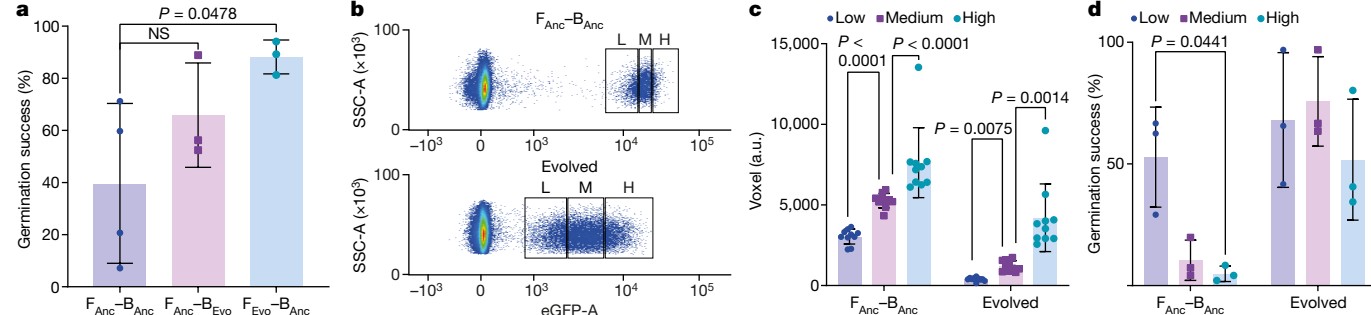

**Fig. 4 | Cross-injection experiments show that the induced endosymbiosis adapted throughout the evolution experiment. a**, The evolved fungus in $F_{Evo}$–$B_{Anc}$ leads to a higher germination success directly after injection compared to the unevolved fungus in $F_{Anc}$–$B_{Anc}$. Data are presented as mean values ± s.d. and compared using two-sided unpaired *t*-test with Welch's correction. *n* = 3 biological samples **b**, Flow cytometry plots of side scatter area (SSC-A) versus enhanced-GFP area (eGFP-A) show a reduced GFP intensity for the bacteria-positive spores from round 10 line 4 (evolved) compared to $F_{Anc}$–$B_{Anc}$ spores. Gates illustrate the approximate gating strategy used to sort samples depicted in **c**,**d**, with the positive population being partitioned into low (L)-, medium (M)- and high (H)-GFP-signal fractions. Prior flow cytometry gating imposed size restrictions and selected for single spores (Methods). **c**, Bacterial load of ten spores per fraction for the low, medium and high fractions collected according to the gating strategy in **b**. There is a significant correlation

between bacterial load and the signal intensity measured with flow cytometry. The bacterial load was determined by quantifying fluorescent voxels of *z*-stack images of single spores. Data are presented as mean values ± s.d. and compared using a two-sided unpaired *t*-test with Welch correction; *n* = 10 technical replicates. *P* = 0.0000000022 (left); *P* = 0.000016 (right). Spores from the evolved pair (round 10 line 4) have a lower bacterial load than $F_{Anc}$–$B_{Anc}$ (all fractions combined, *P* = 0.00000013, two-sided unpaired *t*-test with Welch correction, *n* = 30 technical replicates). **d**, A lower bacterial load correlates with a higher germination success for the ancestral pair ($F_{Anc}$–$B_{Anc}$). This correlation is not detected in the evolved pair, which overall has a lower bacterial load and higher germination success. Data are presented as mean values ± s.d. and compared using two-sided paired *t*-test with individual variance and two-stage set-up. *n* = 3 biological samples.

## Verifying fitness in evolved symbiosis

To verify that the marked increase in fitness of the induced endosymbiosis was due to adaptive evolution, and to test whether one or both partners adapted, we carried out a crossover experiment. For the ancestral partners, we used the same starting stocks as for injection to commence the evolution experiment, and for the evolved partners, we chose the best-performing line after ten rounds of propagation (line 4). We isolated the bacterium from line 4 at the end of the evolution ($B_{Evo}$) and reinjected it into the ancestral fungus ($F_{Anc}$), and reciprocally injected the ancestral bacterium ($B_{Anc}$) into the evolved fungus ($F_{Evo}$; using a bacterium-free spore from the tenth round of line 4). In the first round directly after injection, both cross-pairings had a high germination success, with $F_{Anc}$–$B_{Evo}$ at 66% ± 17% and $F_{Evo}$–$B_{Anc}$ at 88% ± 5%. Thus, the germination success of $F_{Evo}$–$B_{Anc}$ was similar to that of the evolved pair (88%) and significantly higher than that of the ancestral pair (39% ± 29%; Fig. 4a). When we determined the positive fraction in the subsequent round, $F_{Evo}$–$B_{Anc}$ exhibited a positive fraction of 51% ± 13% (Extended Data Table 2). This is even higher than that of the evolved pair (29%), indicative of high fitness and suggesting the possibility of genomic adaptation of the fungus.

In addition to the increases in the positive fraction and germination success throughout the evolution experiment, we observed a lower intensity GFP signal in positive spores from the evolved pair versus the ancestral pair (Fig. 4b). To determine whether the evolved spores contained fewer bacteria, or the bacteria present were expressing less GFP, we correlated the GFP signal obtained from FACS with bacterial load by collecting subpopulations of the positive population (Fig. 4b) and microscopically assessed the bacterial volume in the spores (Methods, Determination of bacterial load). Spores from lower-intensity gates had a lower bacterial volume than spores from higher-intensity gates (Fig. 4c and Extended Data Fig. 2), correlating high GFP intensity as measured by FACS with a higher bacterial load. To evaluate whether the bacterial load influences the ability of the spores to germinate, we determined the germination success of these populations (Fig. 4d). In the ancestral pair, we found that spores from the fraction with a lower bacterial load had a higher germination success than the spores with a high bacterial load. In the evolved pair, all spores had a relatively

low bacterial load compared to the ancestral pair. Comparing spores from the evolved pair with different levels of bacterial load, we still found a trend, but no significant correlation between bacterial load and germination success (Fig. 4d). These findings suggest that a high bacterial load reduces the germination capacity of spores and that the induced endosymbiosis adapted to counteract this issue by reducing the bacterial load.

To test how the presence of bacteria affected the germination capacity of spores as they age, we examined the germination periodically over the course of 27 days. The germination success of spores devoid of bacteria remained relatively constant during this time frame (Extended Data Fig. 3). By contrast, the germination success of bacteria-containing spores for both $F_{Anc}$–$B_{Anc}$ and the evolved pair decreased substantially over time from more than 50% to below 5% after about a month, suggesting that the presence of bacteria takes a continuous toll on the spores.

## Genetic changes of symbiotic partners

Next we analysed whether the phenotypic adaptations in the evolution experiment correlated with genetic changes in the evolved populations. We carried out de novo hybrid genome assembly using both long- and short-read sequencing data to assemble a reference genome for strain NH from the start of the adaptive evolution experiment. Short-read sequencing was used to track changes at several rounds of the experiment (Fig. 3a) for both the fungus and the bacteria. In the bacteria, we did not detect any mutational changes throughout the evolution experiment. In the fungus, we identified a total of 9 different mutations in four different lines that had an allele frequency of at least 50% (Extended Data Fig. 4a). Four mutations were detected in the fittest line, line 4, in which we tracked genetic changes for each passage. The relative frequency of reads carrying the mutation increased sharply towards the end of the experiment and coincided with the increase in the fitness index observed in line 4 (Fig. 5a). We further examined whether the high fitness of the $F_{Evo}$–$B_{Anc}$ pairings correlated with the identified mutations and found all four mutations in all three injected $F_{Evo}$ germlings. In two of three germlings, the mutations occurred with a high frequency of 85–100% of reads and in one germling the frequency was considerably lower with only

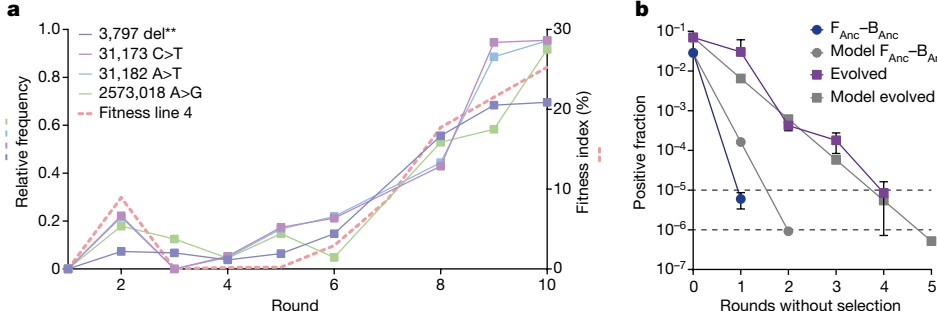

**Fig. 5 | The increased fitness and stability of the evolved endosymbiosis correlates with genetic changes in the fungal population. a**, Four mutations in line 4 became prevalent in the host population during the adaptive evolution experiment, correlating with the increase in the fitness index. The graph depicts in solid lines on the left $y$ axis the relative frequency of reads with the corresponding mutations compared to the reference sequence. The dashed line on the right $y$ axis depicts the fitness index of line 4 over the course of the experiment. Mutations reaching more than 50% relative frequency are shown. 'del**' indicates deletion of two base pairs (CGG>C). **b**, The evolved endosymbiosis is more stable than the ancestral endosymbiosis ($F_{Anc}$–$B_{Anc}$) in

the absence of artificial selection. The graph shows in colour the fraction of bacteria-positive spores in the absence of artificial selection by FACS, and in grey the predicted positive fraction based on a mathematical model. The model uses the germination success and fraction of bacteria-positive and bacteria-negative spores to predict the fraction of bacteria-positive spores over rounds of propagation in the absence of selection. The upper dotted line indicates the threshold for propagation and diluting out (1/100,000); the lower dotted line indicates the detection limit (1/1,000,000). Values measured as 0 positive fraction are not shown on the logarithmic plot. Data are presented as mean values ± s.d. of $n$ = 3 biological samples.

18–35% of reads (note that asexual spores of *Rhizopus* are multi-nucleate[44]). However, in that germling another mutation was found in 61% of reads (Extended Data Fig. 4b). This fifth mutation could be detected in the line 4 population from round 6 onwards, but did not reach the frequency threshold of 50% (Extended Data Table 3). The function of the targeted genes remains speculative. Annotations with InterPro[45] and interaction predictions with STRING[46] suggest that they might be linked to endocytosis, whereas in other lines transcription, translation, mitochondrial maintenance and ion transport may have been affected (Extended Data Table 4). Together, the findings of the sequence analysis confirmed that genomic changes occurred during the adaptive evolution experiment on the host side rather than in the endosymbiont.

## Stability of evolved endosymbiosis

In the adaptive laboratory evolution experiment described above, we exerted a strict positive selection pressure using FACS. In nature, the selection pressure could stem from the bacterially transferred metabolic traits, in this case rhizoxin production[26,31]. However, our initial characterizations of the induced endosymbiosis have also shown substantial costs associated with bacterial occupation, which led to an initially low fitness of the endosymbiosis (Fig. 3d). We therefore reasoned that the induced endosymbiosis would be quickly lost in the absence of positive selection, but would persist for longer after adaptation and mitigation of some of the fitness costs.

To test the stability of the endosymbiosis, we experimentally measured the positive fraction over five rounds of propagation without positive selection and compared the data to a simple mathematical model predicting the fraction of bacteria-positive spores over time. Parameters for the mathematical model were based on measured values of the starting point, namely, the germination success and the initial positive fraction, which we considered to be the most determining factors. For the experiment, positively sorted spores were plated according to the standard conditions in the evolution experiment. Thereafter, 100,000 spores were collected for further cultivation without selecting for bacterial colonization, while simultaneously measuring the positive fraction (Fig. 5b). With the ancestral pair, positive spores diluted out within a single round to below the threshold for propagation (<1/100,000) but were still detectable (>1/1,000,000), which was even faster than the model predicted. For the evolved pair, however, the endosymbiosis was maintained for a longer time as positive spores

diluted out after four to five rounds at the rate that was predicted by the model. This supports the earlier finding that adaptive evolution increased the stability of the induced endosymbiosis, and that the positive fraction and germination success are valid predictors of the fitness of the system.

## Discussion

The emergence of new endosymbioses remains a challenge to observe and study. We have developed an experimental system that allows real-time investigation of the initial steps in the association of a fungal host with an intracellular bacterium. Such encounters are thought to occur frequently in nature, but will be predominantly unstable and transient[3,4,11,47]. The direct implantation allows monitoring of the immediate reactions of pairs with varying degrees of preadaptation, and decoupled from hyphal entry.

The implantation of *E. coli* into *R. microsporus* strains EH and NH resulted in the collapse of the system in a single fungal generation, underscoring the expected instability of new endosymbiotic pairings[4]. Although *E. coli* rapidly divided in the cytosol (Fig. 2a), it was unable to colonize the spores of *R. microsporus*, which represents an evolutionary dead end for endosymbiogenesis. The fungus entrapped densely growing *E. coli* through induced septum formation (Fig. 2a and Supplementary Video 5), probably due to recognition by its innate immune system, which is not well understood at present[42,48]. However, *E. coli* at low densities were not consistently sealed off, suggesting that *R. microsporus* lacks the specificity to clear low-abundance bacteria in its cytosol. This observation implies that *E. coli* might be more likely to be vertically transmitted if it grew slower[36] and/or spread throughout the entire mycelium. The pairing illustrates the necessity for sufficient preadaptation for endosymbiogenesis, as otherwise natural selection cannot begin to take effect.

By contrast, *M. rhizoxinica* implanted into the non-host *R. microsporus* strain NH propagated and dispersed in the fungal hyphae without apparent harm to the fungus and without inducing septum formation (Fig. 2b and Supplementary Video 7). The bacterium reached a subset of spores (Fig. 2c,d), some of which germinated, allowing vertical inheritance (Fig. 2e). We speculate that the bacteria are actively transported along the microtubules, potentially by hitchhiking vesicle transport[41]. This might help the bacteria to spread in the mycelium (Supplementary Videos 2, 3 and 8) and to reach sites where spores form. The colonization of spores in strain NH was notable, because this fungal strain has

not evolved to couple spore formation to bacterial presence[26]. Despite the demonstrated vertical inheritance, the endosymbiosis is rapidly lost in the absence of positive selection (Fig. 5b), probably owing to competition by bacteria-free spores and the observed costs of delayed germination and reduced germination success of colonized spores (Fig. 3b,e). These results are consistent with resource exploitation by the endosymbiont, an effect previously observed in insects[17,18].

When subjected to a strong selection pressure using FACS, the artificially induced endosymbiosis became successively more stable, as evidenced by the slower loss of the endosymbiont when selection pressure was lifted (Fig. 5b). Two major phenotypic factors contributing to this enhancement were an increased fraction of positive spores and 'recovery' of the compromised germination success (Fig. 3b,c). Our cross-validation implantations suggested that adaptation of the fungus was the main contributor to stabilization. This finding was substantiated by genome analyses, which showed enriched mutations in the fungal population but not the bacterial one (Fig. 5a and Extended Data Fig. 4a). This observation is in contrast to findings made in insects, in which the bacterial genome adapts faster[49]. In the fungal coenocytic system, each nuclear duplication can increase host diversity, possibly allowing the fungus to adapt rapidly.

The underexplored genome of this basal fungus makes it difficult to draw firm conclusions about which molecular mechanisms were critical to adapt to the induced endosymbiosis, and additional epigenetic effects on either the host or endosymbiont side cannot be ruled out. Furthermore, genetic changes based on rearrangements or larger deletions or duplications may be missed by short-read sequencing. However, several genes affected by mutations may be involved in the regulation of transcription and translation, and in endocytosis (Extended Data Table 4). Although the exact mechanisms remain unclear, we could show phenotypically that the system evolved to increase the number of spores with the bacterium, and we identified reduced bacterial load in the spores as an important factor for the increased germination success (Fig. 4b–d).

Our findings indicate that *M. rhizoxinica* is largely compatible with non-host *R. microsporus* strain NH, but maintaining the endosymbiont requires selection pressure. In the natural endosymbiotic pair, the costs for harbouring the bacterium are probably offset by benefits to the fungus, such as co-production of rhizoxin[23,31]. We found that the implanted bacterium produced WF-1360F within *R. microsporus* strain NH. We also detected rhizoxin, which is produced when *R. microsporus* epoxidizes WF-1360F congeners[29], indicating that the ability to modify WF-1360F is not exclusively present in *R. microsporus* strains that evolved to harbour *M. rhizoxinica*. The production of rhizoxin by strain NH demonstrates that metabolic capabilities can be transferred to new organisms by artificially inducing an endosymbiosis through implantation and subsequent selection.

In summary, we have successfully adopted the FluidFM technology to implant live bacteria into a fungus. This single-cell approach enables real-time imaging of ensuing interactions and to study the prerequisites for endosymbiogenesis even in short-lived and unstable pairings. Combined with the high-throughput nature and sensitivity of FACS, rare vertical transmission events can be analysed and cultured under strict selection. We found that adaptive evolution increased the stability of the induced endosymbiosis despite high initial costs. The ability to initiate endosymbioses and to explore the boundary conditions for stable inheritance advances synthetic approaches towards biotechnologically relevant designer endosymbionts and sheds light on the evolutionary forces shaping endosymbiogenesis.

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

## Methods

### Strains and standard culturing methods

The strains and chemicals used are listed in Supplementary Tables 1 and 2. *E. coli* K-12 BW25113 pDGUV-GFP[50] was cultured in Luria–Bertani broth (Sigma-Aldrich) supplemented with 100 µg ml⁻¹ carbenicillin (Roth). *R. microsporus* strain NH is *R. microsporus* CBS 631.82; *R. microsporus* strain EH is *R. microsporus* ATCC62417. *M. rhizoxinica* HKI-0454 and *R. microsporus* were cultured as previously reported[42] at 28 °C. *M. rhizoxinica* was fluorescently labelled with cytosolic GFP using pBBR-P12-GFP[51]. *R. microsporus* was grown on potato dextrose agar (ThermoFisher) at 28 °C with 20 µg ml⁻¹ gentamicin. Respective antibiotics for plasmid retention were used as required for the culture of bacteria in all experiments, including the growth of bacteria inside the fungus, in which case, additionally, gentamicin was used to prevent extracellular bacterial growth.

### Bacterial injections with FluidFM

The basic setup of the instruments, FluidFM probe processing steps, probe cleaning and probe coating were as previously described[36], but the pressure was controlled with a FlowEZ 7000 (Fluigent), in the range of 0–7,000 mbar, using an AT550-9L compressor (WenLing) as pressure source. The probe apex was sharpened with a Helios 5UX DualBeam focused ion beam scanning electron microscope (ThermoFisher) as described previously[36,52], but to a new shape that resulted in a sharp double point, as shown in Fig. 1, by targeting the probe from the front and shaping a centred point with a 60° central angle. The cantilevers of the used probes had a nominal stiffness of 1.6 N m⁻¹. The fungal sample was prepared by seeding spores in 50-mm WillCo glass-bottom dishes (WillCo Well) and adding 4 ml potato dextrose broth (PDB) medium (ThermoFisher) containing 34 µg ml⁻¹ chloramphenicol, incubating at room temperature for 14–16 h, washing 1–3 times with 4 ml PDB, depending on germling concentration, exchanging the medium for protoplasting mix (to soften but not completely disintegrate the cell wall; 1.6 g Cellulase Onozuka R10 (Duchefa Biochemie), 40 mg chitinase (Merck), 40 ml iced MMB (0.5 M mannitol, 0.05 M maleate pH 5.5), filtered through a 0.22-µm syringe filter), and incubating for another 3–5 h at room temperature. If germlings were too dense, the sample supernatant was filtered through a 0.22-µm syringe filter. The *E. coli* sample was grown in a 10-ml culture in a baffled shake flask overnight at 37 °C, washed three times in Hepes2 buffer (10 mM HEPES, 150 mM NaCl, pH 7.4), and adjusted to an optical density of 2 at 600 nm. The *M. rhizoxinica* sample was grown in a 2-ml culture in a 12-ml culturing tube for 3 to 5 days at 28 °C and prepared like the *E. coli* culture. Bacterial suspensions (15 µl) were pipetted into the reservoir of the FluidFM probe. The probe was moved towards the glass surface ($z = 0$ µm) beside the targeted germling and then retracted to a $z$-value of +8 to +10 µm. The injection site of the germling was chosen on the basis of direct proximity to the glass surface, preferring thick germtube portions. As force-controlled puncturing was not possible owing to the softness of the probe, the germling was punctured by advancing the probe by 10 µm (corresponding to a nominal $z$-value of +1 µm to −2 µm), holding this position for 5 s and retracting to a $z$-value of +1 to +3 µm. Successful puncturing of the germling could be monitored with wide-field illumination by observing cytoplasmic flow. Immediately following puncture of the germling, the microfluidic system was pressurized with 3–4 bar, stopping turgor-induced backflow into the probe. Subsequently, the pressure was increased to 6.5 bar until liquid flow into the germling was noticed, and then quickly reduced to 3–4 bar to prevent bursting of the germling. Injection of bacteria was confirmed by switching to the fluorescence channel, and then the pressure was slowly reduced to 0 mbar to allow recovery of the germling. The injected germling was isolated to a fresh dish filled with recovery medium (3.8 ml MMB, 1 ml PDB, 160 µl 4 M sorbitol) after 3–10 min, or after the germling recommenced growth. Isolation was carried out by lifting the FluidFM and exchanging the sample dish for the recovery dish, with the germling sticking to the probe. Recovery of the germling and the ensuing dynamics were visualized using time-lapse and $z$-stack images in the wide-field and the fluorescence channels. Some of the injections were carried out with the addition of 1:1,000 calcofluor white (Merck) in the protoplasting mix and recovery medium to visualize the cell wall. Germlings were grown in recovery medium overnight and then detached from the probe using overpressure, lifting of the FluidFM and scraping with a plastic pipette tip. Mycelium was then transferred to a potato dextrose agar plate and incubated at 28 °C.

### Spore collection

Spores were collected 6 ± 1 days after injection or plating from spores. Spore solution (8.5% NaCl, 1% Tween 20) was added to plates (16 ml for square plates; 12 ml for round plates), and the spores were thoroughly detached using a spatula. The remaining mycelium was clumped up and gently pressed with the spatula to release the spore solution from the mycelium. The spore solution was filtered through a 10-µm CellTrics filter (Sysmex). Spores were washed three times with 1 ml Hepes2 (relative centrifugal force of 8,000; 2 min) and stored at 4 °C overnight for FACS, or in 50% glycerol at −20 °C (for working stocks) or −80 °C (for long-term storage).

### Flow cytometry and cell sorting

Analysis and sorting of spores were conducted at the ETH Flow Cytometry Core Facility on a FACSAria Fusion BSL2 cell sorter (BD). Single spores were selected using SSC-A–FSC-A and FSC-H–FSC-A gates. Colonization by bacteria was checked using an SSC-A–eGFP-A gate. (See Supplementary Fig. 1 for the gating strategy). For samples for which autofluorescence was suspected to complicate the positioning of the positive gate, an mCherry-A versus PerCP–Cy5-5-A channel was used to check for autofluorescence but was not included in the gating. For determination of the fraction of positive spores, 100,000 to 1,000,000 spores were analysed depending on the size of the fraction. For bulk sorting, spores were sorted into 1.5-ml Eppendorf tubes. For determination of germination success, single spores were sorted into 96-well plates containing 125 µl PDB + 34 µg ml⁻¹ chloramphenicol per well. For verification of positive gates, bulk-sorted spores were intermittently checked under the microscope. Collected data were analysed using FlowJo v10 software (BD). For sorting of high, medium and low fractions of positive spores, gates were set qualitatively in the positive population as illustrated in Fig. 4b.

### Determination of germination success

Single-spore-sorted 96-well plates were incubated at 28 °C and checked visually with a Zeiss SteREO Discovery.V8 microscope (Zeiss) for the appearance of a germling. Per sample, three plates of positive spores and one plate of negative spores were sorted. In round 1 of the adaptive laboratory evolution experiment, five positive plates were sorted per sample. Germlings were counted 1 day and 2 days after sorting, after which point there are no new germlings to be discovered. For positive plates, five germlings were checked microscopically on day 1 to confirm the presence of fluorescent bacteria. If a germling could not be confirmed to have endobacteria, five more germlings were checked. No samples failed this control by having more than 20% of germlings without easily detected endobacteria. To calculate the percentage of delayed germinations, the percentage of germlings not detected on day 1 but detected on day 2 was calculated (with 100% corresponding to the number of germlings detected on day 2).

### Bacterial isolation

For isolation of *M. rhizoxinica* from *R. microsporus*, 50 ml MGYM9 + 34 µg ml⁻¹ chloramphenicol was inoculated with spores in a 500-ml baffled shake flask and incubated with shaking at 100 r.p.m. for 5–7 days. Build-up of mycelium on the wall was periodically flushed off

by gentle shaking and tilting by hand. Once the medium became turbid, an inoculation loop was inserted in the medium, avoiding mycelial clumps and used for streaking out on agar plates. Plates were checked daily for 3 days and growing fungus was cut out if detected. Plates with bacterial colonies were further cultivated according to standard protocols and were used to make cryogenic stocks. Alternatively, a 2-μm syringe filter (Merck) was used to separate bacteria from mycelium and spores, and the filtrate was used for further cultivation in liquid.

### Adaptive laboratory evolution experiment

For the evolution experiment, spores collected in the first round after injection were sorted for determination of germination success, and the remaining spores were bulk sorted for positive spores. The positive spores were split into 10 equal lines, resulting in approximately 300 spores per line. Thereafter, the ten lines were kept separate. The germination success and the fraction of positive spores were determined in every round for every line, and at every round after round 1, some positively and negatively sorted spores were cryopreserved, and bacteria were isolated from 10,000 positively sorted spores (approximately 300 for round 1). The product of the positive fraction and the germination success was calculated to give the fitness index, indicating the percentage of spores that would yield bacteria-populated germlings for the start of a next round without selection. Plating for the evolution experiment was carried out on 120 × 120 mm square Petri dishes (Greiner) to increase the available surface area. Spores to be plated were taken up in 100 μl of buffer, which was pipetted into five parallel lines with equal spacing on the plate. Standard spread plating for high spore numbers had previously proved to lead to inconsistent spore formation. From round 2 on, high densities of spores could be plated for the next round as the fraction of positive spores increased. For making round 3 plates, the highest number of spores was seeded, up to 830,000 spores per plate, whereas for rounds 4 and 5 the numbers varied slightly around 100,000, and from thereon 100,000 were used for all rounds and lines. The number of seeded spores is shown in the Source Data for Fig. 3a. After round 7, only the three best-performing lines according to the fitness index were grown individually (lines 2, 4 and 7), whereas the other seven lines were pooled to line P by mixing equal amounts of positive spores. A scheme indicating the population sizes of spores and the pooling regime can be found in Fig. 3a.

### Detection of rhizoxin

A total of 10,000 positively sorted spores per line from round 7 were plated. Strain EH colonized by *M. rhizoxinica*, the axenic strain NH and a liquid culture of *M. rhizoxinica* served as controls. Plates were grown for 11 days and extracted with 50 ml ethyl acetate with shaking at 100 r.p.m. overnight at 28 °C. The organic phase was separated, dried with sodium sulfate, filtered through paper filter, and evaporated with a rotary vacuum evaporator. Samples were taken up in 1 ml acetonitrile, centrifuged at 20,000*g* for 10 min and 800 μl of the supernatant was stored at −80 °C until liquid chromatography with tandem mass spectrometry (LC–MS/MS) analysis. LC separation was carried out with a Thermo Ultimate 3000 UHPLC system (Thermo Scientific) using a C18 reversed-phase column (Kinetex XB-C18 column, particle size 1.7 μm, pore size 100 Å; dimensions 50 mm × 2.1 mm, Phenomenex). Solvent A was 0.1% (v/v) formic acid in water and solvent B was 0.1% formic acid in acetonitrile at a flow rate of 500 μl min$^{-1}$. Solvent B was varied as follows: 0 min, 25%; 3 min, 90%; 5 min, 90%; 5.3 min, 25%; subsequently, the column was equilibrated for 2 min at the initial condition. The injection volume was 2 μl.

MS-product reaction monitoring analysis was carried out with a Thermo QExactive plus instrument (Thermo Fisher Scientific) in the positive Fourier transform mass spectrometry mode. MS level 1 scans were carried out with a mass resolution of 35,000 ($m/z = 200$) and MS level 2 scans were carried out with a mass resolution of 17,500. Parent ions were isolated at $m/z$ 594.34, 610.337, 612.3531, 626.3323 and 628.348 with unit resolution and fragmented by high-energy C-trap collision dissociation applying a normalized collision energy of 28 eV. A heated electrospray ionization probe was used with the following source parameters: vaporizer temperature, 380 °C; sheath gas, 50; auxiliary gas, 20; sweep gas, 0; RF level, 50.0; capillary temperature, 275 °C. See Supplementary Fig. 2 for relevant spectra.

### Genomics

For the generation of sequencing samples of *M. rhizoxinica*, bacteria were grown according to standard culturing conditions after isolation from the respective time point (Methods, Bacterial isolation), and 4 ml of sample adjusted to an optical density of 1 at 600 nm were pelleted by spinning at a relative centrifugal force of 11,000 for 1 min. Genomic DNA was prepared using the MasterPure DNA Purification Kit (LGC). Genomic DNA was sent on dry ice to BMKGene (Biomarker Technologies) for further processing.

For generation of sequencing samples of *R. microsporus*, mycelium was grown in 500 ml malt extract broth (Thermo Fisher Scientific) in 2-l shake flasks for Illumina sequencing or in 1.5-l malt extract broth in 5-l shake flasks for PacBio sequencing at 37 °C for 5 days, with the addition of gentamycin and chloramphenicol. The mycelium was then filtered on a 110-mm filter paper and washed thoroughly with double-distilled H$_2$O and for Illumina samples additionally with 150 ml of 70% ethanol. The mycelium was then removed from the filter paper, packed into a 50- ml screw cap tube and frozen in liquid nitrogen. The samples were then sent to BMKGene (Biomarker Technologies) for further processing.

For genome assembly of *R. microsporus* CBS 631.82, Pacbio HiFi sequencing with RS II produced a total of 3,867,257,442 base pairs (bp) in 353,300 reads of average length 10.9 kilobases and average quality score 30.3. The reads were assembled with Flye (v2.9.2)[53] with the --pacbio-hifi flag, resulting in 118 contigs of total length 55,743,399 bp with an N50 (the shortest contig of the set of the largest contigs making up 50% of the assembly) of 1,370,944 bp. BUSCO (v5.4.7)[54] was used to check the quality of the assembly, using the lineage dataset mucorales_odb10, giving the following result: C (complete): 97.5%, S (single-copy): 5.1%, D (duplicated): 92.4%, F (fragmented): 1.7%, M (missing): 0.8%, *n* (number of genes): 2,449. Thus, although nearly complete, the genome also seems to be largely duplicated. The assembly was gene-called with BRAKER (v3.0.6)[55–62], using the −fungus flag, and then functionally annotated with eggNOG-mapper (v2.1.12)[63] using the option --target_taxa Fungi. BUSCO reported a slightly improved completeness for the called genes: C: 99.8% [S: 1.4%, D: 98.4%], F: 0.1%, M: 0.1%, n: 2,449.

For calling mutations from the adaptive evolution experiment for *R. microsporus* and *M. rhizoxinica*, the short-read Illumina sequences provided by BMK were used. The resulting raw reads were cleaned by removing adaptor sequences, low-quality-end trimming and removal of low-quality reads using BBTools v38.18 (https://source-forge.net/projects/bbmap/). The commands used for quality control can be found on the Methods in Microbiomics web page (https://methods-in-microbiomics.readthedocs.io/en/latest/preprocessing/preprocessing.html). Single nucleotide polymorphisms were called using two different tools—Snippy and bcftools[64]. For variant-calling with bcftools, reads were first aligned to the PacBio assembly of *R. microsporus* CBS 631.82 or to the *M. rhizoxinica* reference genome (GCF_000198775.1) using BWA-MEM v0.7 (ref. 65). Duplicate reads were marked and removed using GATK4 v4.2 (MarkDuplicates)[66]. Variants were called using the bcftools call command. Single nucleotide polymorphism calls were validated by comparing results obtained by two independent tools. All the variants that were detected in the ancestral sample were filtered out from all of the evolved samples using bcftools-isec to investigate variants that arose during the evolution experiment. The resulting VCF files were filtered using bcftools with the following criteria: -Ov -sLowQual -g5 -G10 -e 'QUAL < 200 || DP4[2] < 3 || DP4[3] < 3 || (DP4[2] + DP4[3])/sum(DP4) < 0.1 || MQ < 50'. Each variant call produced by each of the tools was manually checked by

analysing the read alignments at variant positions using samtools pileup. Mutations were annotated using SnpEff[67] and InterProScan[45]. STRING was used to identify putative biological processes by searching for interactions on the basis of the closest-related genes for fungi found in the STRING database[46].

### Fitness without selection pressure

Spores for round 0 were first collected from a plate grown according to the standard conditions from the adaptive laboratory evolution experiment. For this, spores from a fresh $F_{Anc}$–$B_{Anc}$ injection plate were bulk sorted and 100,000 positive spores were plated. For $F_{Evo}$–$B_{Anc}$, frozen positive spores from an injection plate were grown and then 100,000 positive spores were plated; for $F_{Evo}$–$B_{Evo}$, frozen spores from the evolution experiment from round 10 line 4 were grown and then 100,000 positive spores were plated. From these plates, spores of round 0 were collected. Then 100,000 spores were bulk sorted gating only for single spores but disregarding the GFP signal intensity of the spores and used for subsequent rounds. Every round, 1,000,000 spores per sample were analysed to determine the fraction of positive spores. The endosymbiosis was considered washed out once the positive fraction fell below the threshold of 1/100,000 spores, at which no positive spore is expected to be plated for the next round. The experiment was stopped after round 5. The theoretical trajectory of the positive fraction was calculated using the following formula:

$$p_x = (p_{x-1} \times g \times p_0)/(p_{x-1} \times g + [1 - p_{x-1}] \times e),$$

where $p$ is the positive fraction; $x$ is the round for which the positive fraction is being determined; $g$ is the germination success of positive spores as measured in round 0; $e$ is the germination success of negative spores derived from the long-term average from the evolution experiment (69%); and $p_0$ is the positive fraction measured in round 0 to describe how many spores formed by bacteria-positive germlings are again bacteria positive.

### Determination of bacterial load

Spores sorted for high, medium and low intensity were imaged in the wide-field and GFP channel to create z-stacks. Overlays of 10 spores per population per sample are shown in Extended Data Fig. 2a,b and served as visual confirmation for the correlation of GFP intensity as measured by FACS and bacterial population size. Contrast and brightness were kept the same. The volume of voxels considered to be bacteria was calculated using Matlab2018a (Mathworks) using sections of published code[36]. To gauge the appropriate threshold, the three-dimensional rendering of the resulting voxel cloud was visually inspected, and the diameter of single particles was checked. An example of the resulting three-dimensional rendering of samples is shown in Extended Data Fig. 2c.

### Spore longevity

Samples were grown as described in the section entitled Fitness without selection pressure. Single spores were sorted into 96-well plates with 75 µl of Hepes2 + 34 µg ml$^{-1}$ chloramphenicol in each well. Per sample, 15 plates with negative spores and 15 plates with positive spores were sorted. Plates were then incubated at 16 °C. One day before the nominal time point, three plates per condition were activated by adding 125 µl of PDB + 34 µg ml$^{-1}$ chloramphenicol to each well and incubating at 28 °C.

### Statistical analyses

Statistical tests indicated in the figure legends were run with GraphPad Prism v9.0.0.

### Images, videos, plots and figures

Images and videos were edited using Fiji[68] for contrast, z-stacks, time stamps, overlays and scale bars, as indicated in the respective sections of the Methods and figure legends. Videos were cut together and labelled using iMovie v10.4 (Apple). Plots were generated using GraphPad Prism 9. Figures were assembled and edited using Adobe Illustrator 2020. Illustrations for Fig. 1 were created with BioRender.com.

### Reporting summary

Further information on research design is available in the Nature Portfolio Reporting Summary linked to this article.

## Data availability

Genomic data are available in the European Nucleotide Archive under the accession number PRJEB76713. For an ID key, see Supplementary Table 3. FACS data and analyses, statistical analyses and plots, LC–MS/MS data and raw images for all figures and Supplementary Videos 1–4 are available on Zenodo at https://doi.org/10.5281/zenodo.12189101 (ref. 69). Raw images for Supplementary Videos 5–8 are available on Zenodo at https://doi.org/10.5281/zenodo.12518583 (ref. 70). The STRING[46] database is available at https://string-db.org/. Source data are provided with this paper.

## Code availability

The code used in this publication is described in the Methods and is available on Zenodo at https://doi.org/10.5281/zenodo.13309607 (ref. 71).

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

**Acknowledgements** This work was supported by a European Research Council Advanced Grant (SYMBIOSES number 883077 to J.A.V.); the National Centre of Competence in Research (NCCR) Microbiomes, financed by the Swiss National Science Foundation (51NF40_180575) to S.S. and J.A.V.; and the Deutsche Forschungsgemeinschaft (German Research Foundation) under Germany's Excellence Strategy—EXC 2051—Project-ID 390713860, the SFB 1127 ChemBioSys—Project-ID 239748522 and the Leibniz Award to C.H. I.R. is grateful for financial support from the European Union's Horizon 2020 Research and Innovation Program under the Marie Skłodowska-Curie grant agreement number 794343. We thank E. Sarajlic, P. Christen, the personnel of the Scientific Center for Optical and Electron Microscopy at ETH Zurich and the personnel of the Flow Cytometry Core Facility at ETH Zurich for support and technical assistance.

**Author contributions** G.H.G. and J.A.V. conceived the project. G.H.G., C.E., I.R., C.G.G., O.G.-G., M.K., C.H. and J.A.V. designed the study. G.H.G., C.E., C.G.G., T.Z., M.K. and O.G.-G. developed the injection protocol. G.H.G. carried out the injections, FACS measurements, adaptation characterization experiments and microscopy and analysed the data. G.H.G., T.G. and M.B.-M. carried out the adaptive laboratory evolution experiment and G.H.G. analysed the data. C.M.F., A.S., G.H.G. and S.S. acquired and analysed the genomics data. P.K. and G.H.G. carried out the LC–MS/MS measurements and P.K., K.S. and G.H.G. analysed the data. G.H.G. and C.G.G. processed FluidFM cantilevers. G.H.G. and J.A.V. wrote the manuscript with input from all authors.

**Funding** Open access funding provided by Swiss Federal Institute of Technology Zurich.

**Competing interests** The authors declare no competing interests.

**Additional information**
**Correspondence and requests for materials** should be addressed to Julia A. Vorholt.

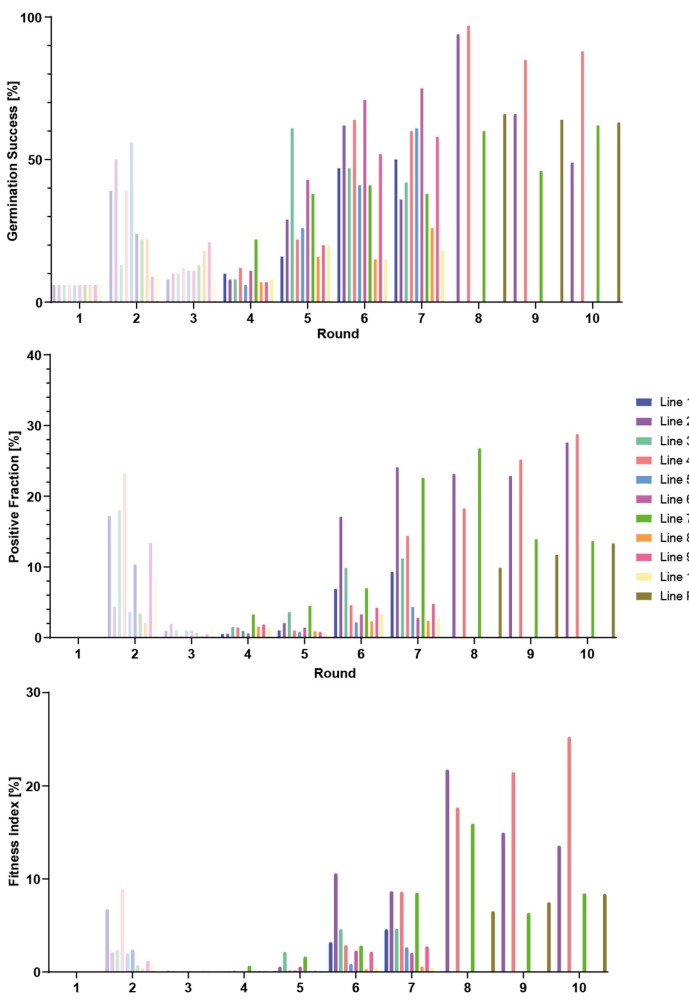

**Extended Data Fig. 1 | Plots showing data for all ten rounds of the adaptive laboratory evolution experiment for all ten lines.** Pale colored datapoints in rounds 1–3 indicate the expansion phase of the experiment under differing plating densities (see Methods and Extended Data Fig. 4a). The increase in fitness in round 2 is likely due to the low-density plating conditions. Plots show the heterogeneity in increase of the fitness of the different lines and trend towards higher fitness. For round 8–10, only lines 2, 4, and 7, were continued as before, while line P was the pooled fraction of the other lines.

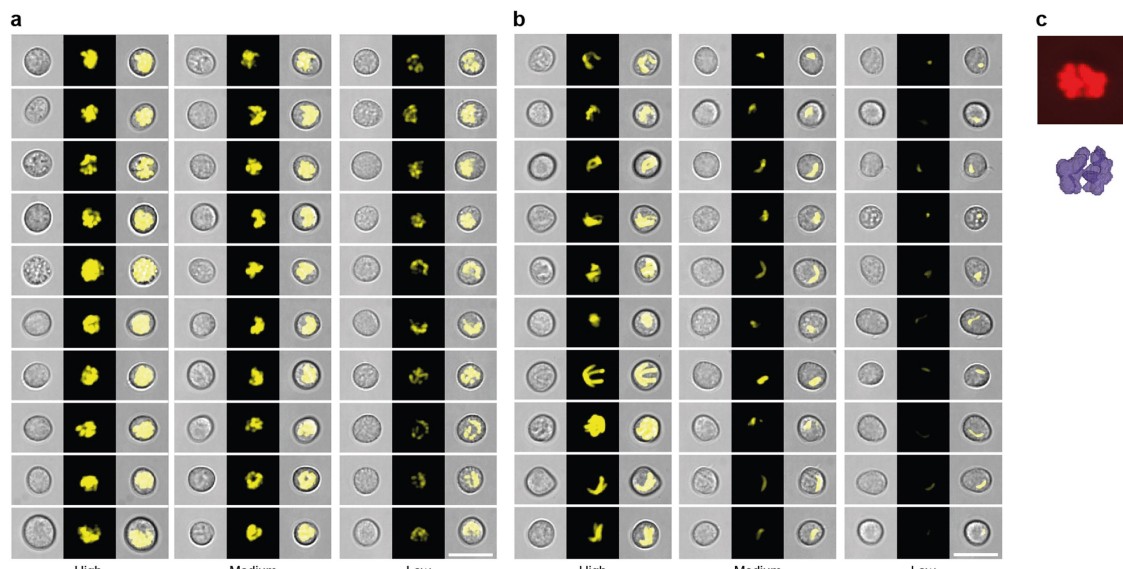

a High Medium Low

b High Medium Low

c

**Extended Data Fig. 2 | Bacterial load assessment inside spores of R. microsporus strain NH.** Panel **a** and **b** show per column from left to right: a single z layer widefield image, a 2D sum of the GFP signal of a z stack in yellow, the overlay of the two signals. The different columns show 10 spores each from FACS-sorted spores from fractions with high, medium, and low GFP signal. Scale bar 10 μm. Panel **a** shows spores from the unevolved partners. Panel **b** shows spores from the evolved partners. The microscopy data shows that lower FACS signal corresponds to a lower bacterial load. Bacteria from the evolved endosymbiosis show an elongated phenotype and generally a lower bacterial load. Panel **c** shows an example of the output for the computational analysis of bacterial volume used for Fig. 5c. In red on black is the fluorescence signal used for the selection of the region of interest (same spore as Panel a, first column, third row). In purple on white is a pseudo 3D rendering of the generated output for visual confirmation. The calculated voxel number for this sample was 6247.

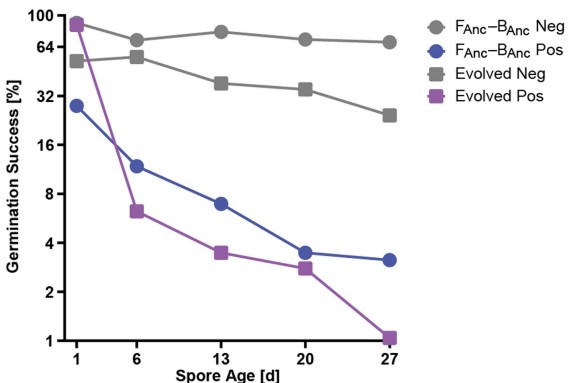

**Extended Data Fig. 3 | The germination rate of bacteria-positive spores drops faster over time than the germination rate of bacteria-negative spores.** The plot shows the germination rate of spores at timepoints after harvesting and incubating in Hepes2 at 16 °C on a logarithmic scale. Evolved spores stem from Round 10 Line 4. Data shown is from a single sample.

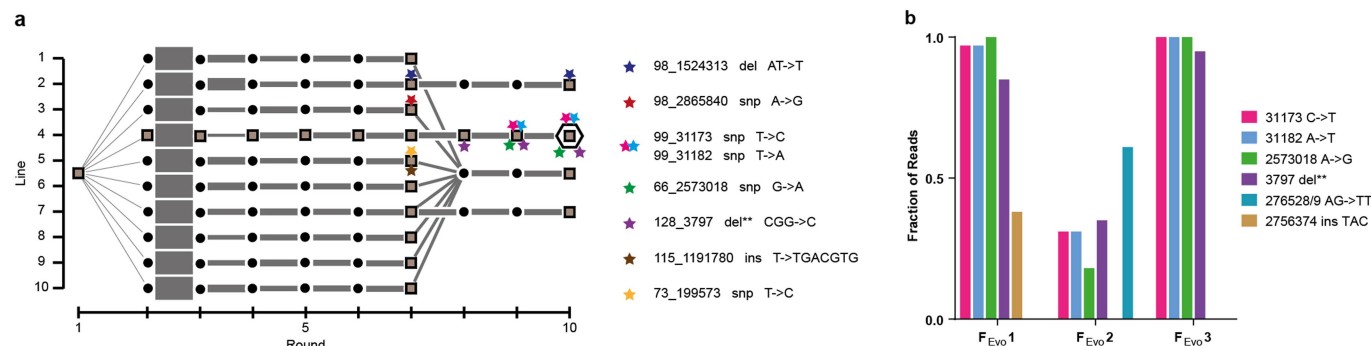

**Extended Data Fig. 4 | Overview of the identified mutations in the adaptive laboratory evolution experiment. a** Scheme of mutations found in over 50% of reads throughout the adaptive evolution experiment (see also Fig. 3a). Squares indicate where genomic DNA was extracted from the fungus and from reisolated bacteria. Colored stars indicate samples where the respective mutations is found in more than 50% of reads. The hexagon marks round 10 of line 4, the population with the highest final fitness index, which was used for phenotypic characterization and is referred to as "evolved" (positive spores), $F_{Evo}$ (negative spores), or $B_{Evo}$ (reisolated bacteria). **b** Graph showing the genotype of the mycelium formed after injecting an $F_{Evo}$ negative spore with $B_{Anc}$ bacteria, which showed a high fitness index. Shown are the fraction of reads showing the mutations for the four loci identified in panel a plus an additional 2 loci which showed differences to the reference in more than 30% of reads. Details to the frequency of these two mutations over the course of the experiment can be found in (Extended Data Table 3) $F_{Evo}1$ and $F_{Evo}3$ appear to almost exclusively have the four previously identified mutations, whereas they are only partially present in $F_{Evo}2$.

**Extended Data Table 1 | The fraction of positive spores in samples of *R. microsporus* strain NH after injection of GFP labelled *M. rhizoxinica* ($F_{Anc}$–$B_{Anc}$)**

| $F_{Anc}$–$B_{Anc}$ | Positive fraction [%] |
|---|---|
| (adaptive Evolution) (Inj4) | 0.01 |
| Repetition 1 (Inj8) | 4.09 |
| Repetition 2 (Inj9) | 0.06 |
| Repetition 3 (Inj10) | 0.009 |

**Extended Data Table 2 | The fraction of positive spores in samples of $F_{Evo}$–$B_{Anc}$ after propagation with standard conditions**

| $F_{Evo}$–$B_{Anc}$ | Positive fraction R2 [%] |
| --- | --- |
| Repetition 1 (TInj1) | 62.8 |
| Repetition 2 (TInj3) | 33.9 |
| Repetition 3 (TInj4) | 57.5 |

**Extended Data Table 3 | The frequencies of two mutations detected in re-injected germlings of $F_{Evo}$–$B_{Anc}$ throughout the evolution experiment in Line 4**

| Round | Frequency of 276528/9 AG->TT | Frequency of 2756374 ins TAC | Fitness L4 [%] |
|---|---|---|---|
| 1 | 0 | 0.011 | 0.001 |
| 2 | 0 | 0.015 | 8.938 |
| 3 | 0 | 0 | 0.041 |
| 4 | 0 | 0 | 0.175 |
| 5 | 0 | 0 | 0.225 |
| 6 | 0.15 | 0 | 2.923 |
| 7 | - | - | 8.65 |
| 8 | 0.25 | 0 | 17.718 |
| 9 | 0.093 | 0.011 | 21.508 |
| 10 | 0.019 | 0 | 25.282 |

**Extended Data Table 4 | Information about the mutations discovered in samples of the evolution experiment**

| Contig | Locus | Type | Ref | Alt | Present in | Relative location and predictions |
|---|---|---|---|---|---|---|
| 66 | 2573018 | snp | G | A | R9L4, R10L4, $F_{Evo}1$, $F_{Evo}3$ (R2L4, R3L4, R4L4, R5L4, R6L4, R7L4, R8L4, $F_{Evo}2$) | Missense in g18079 (+)<br>InterPro: None; none (mostly disordered)<br>STRING: Clathrin dependent endocytosis |
| 71 | 276528 | snp | A | T | $F_{Evo}3$ (R6L4, R8L4, R9L4, R10L4) | Between ends of g19929 (527 nt) and g19930 (463 nt)<br>InterPro g19929: Transposase IS630-like; DNA transposition, DNA integration<br>InterPro g19930: None, DNA transposition, DNA integration |
| 71 | 276529 | snp | G | T | $F_{Evo}3$ (R6L4, R8L4, R9L4, R10L4) | STRING g19929 and g19930: delta(3)-delta(2)-enoyl-CoA isomerase activity (gene ontology) |
| 73 | 199573 | snp | T | C | R7L5 (R10LP) | In intron of g20401<br>InterPro: Maintenance of mitochondrial morphology protein 1; mitochondrion organization<br>STRING: Mitochondrial genome maintenance |
| 98 | 1524313 | del | AT | A | R7L2, R10L2 (noise) | In intron of g23606<br>InterPro: FACT complex subunit Spt16; none (facilitates chromatin transcription complex)<br>STRING: recruitment of 3'-end processing factors to RNA polymerase II holoenzyme complex |
| 98 | 2756374 | ins |  | TAC | ($F_{Evo}1$) | In exon (frame shift) of g24179<br>Interpro: None; none<br>STRING: Translation |
| 98 | 2865840 | snp | A | G | R7L3 (R10LP) | In intron of g24232<br>InterPro: P-type ATPase, subfamily IIIA; proton export across plasma membrane<br>STRING: Inorganic cation transmembrane transport |
| 99<br>99 | 31173<br>31182 | snp<br>snp | T<br>T | C<br>A | R9L4, R10L4, $F_{Evo}1$, $F_{Evo}3$ (R2L4, R4L4, R5L4, R6L4, R7L4, R8L4, $F_{Evo}2$) | 60/69 nt before g24285<br>InterPro: None; none (complete protein disordered)<br>STRING: None |
| 115 | 1191780 | ins | T | TGACGTG | R7L5 (R10LP) | 2195 nt before g2811<br>InterPro: Glutathione transferase family; translational elongation<br>STRING: Translation |
| 128 | 3797 | del | CGG | C | R10L4, $F_{Evo}1$, $F_{Evo}3$ (R2L4, R3L4, R4L4, R5L4, R6L4, R7L4, R8L4, R9L4, R2L10, $F_{Evo}2$) | 1617 nt before g6615<br>InterPro: None; proteolysis<br>STRING: None |

# Reporting Summary

## Statistics

For all statistical analyses, confirm that the following items are present in the figure legend, table legend, main text, or Methods section.

| n/a | Confirmed | |
|---|---|---|
| ☐ | ☒ | The exact sample size (*n*) for each experimental group/condition, given as a discrete number and unit of measurement |
| ☐ | ☒ | A statement on whether measurements were taken from distinct samples or whether the same sample was measured repeatedly |
| ☐ | ☒ | The statistical test(s) used AND whether they are one- or two-sided *Only common tests should be described solely by name; describe more complex techniques in the Methods section.* |
| ☒ | ☐ | A description of all covariates tested |
| ☒ | ☐ | A description of any assumptions or corrections, such as tests of normality and adjustment for multiple comparisons |
| ☐ | ☒ | A full description of the statistical parameters including central tendency (e.g. means) or other basic estimates (e.g. regression coefficient) AND variation (e.g. standard deviation) or associated estimates of uncertainty (e.g. confidence intervals) |
| ☐ | ☒ | For null hypothesis testing, the test statistic (e.g. *F*, *t*, *r*) with confidence intervals, effect sizes, degrees of freedom and *P* value noted *Give P values as exact values whenever suitable.* |
| ☒ | ☐ | For Bayesian analysis, information on the choice of priors and Markov chain Monte Carlo settings |
| ☒ | ☐ | For hierarchical and complex designs, identification of the appropriate level for tests and full reporting of outcomes |
| ☒ | ☐ | Estimates of effect sizes (e.g. Cohen's *d*, Pearson's *r*), indicating how they were calculated |

*Our web collection on statistics for biologists contains articles on many of the points above.*

## Software and code

Policy information about availability of computer code

| Data collection | Data collection as described in the methods section, including VisiView v4.4.0.8 |
|---|---|
| Data analysis | Data analysis as described in the methods section, including FlowJo v10, Flye v2.9.2, BUSCO v5.4.7, BRAKER v3.0.6, BBTools v 38.18, Snippy, bcftools, bcftools-isec, BWA-MEM v0.7, GATK4 v4.2, SnpEff, InterProScan, STRING, Matlab2018a, GraphPad Prism v9.0.0, Fiji 2, iMovie v3.0, GraphPad Prism 9, Adobe Illustrator 2020, BioRender.com and the STRING database: https://string-db.org/. Collected data available under 10.5281/zenodo.12189101, 10.5281/zenodo.12518583, 10.5281/zenodo.13309607, and PRJEB76713 (ENA). |

For manuscripts utilizing custom algorithms or software that are central to the research but not yet described in published literature, software must be made available to editors and reviewers. We strongly encourage code deposition in a community repository (e.g. GitHub). See the Nature Portfolio guidelines for submitting code & software for further information.

## Data

Policy information about availability of data

All manuscripts must include a data availability statement. This statement should provide the following information, where applicable:

- Accession codes, unique identifiers, or web links for publicly available datasets
- A description of any restrictions on data availability
- For clinical datasets or third party data, please ensure that the statement adheres to our policy

The data availability statement contains the DOIs for the repositories where the data is stored. The repositories will be made public before publication.

## Research involving human participants, their data, or biological material

Policy information about studies with human participants or human data. See also policy information about sex, gender (identity/presentation), and sexual orientation and race, ethnicity and racism.

| | |
|---|---|
| Reporting on sex and gender | N/A |
| Reporting on race, ethnicity, or other socially relevant groupings | N/A |
| Population characteristics | N/A |
| Recruitment | N/A |
| Ethics oversight | N/A |

Note that full information on the approval of the study protocol must also be provided in the manuscript.

# Field-specific reporting

Please select the one below that is the best fit for your research. If you are not sure, read the appropriate sections before making your selection.

☒ Life sciences        ☐ Behavioural & social sciences        ☐ Ecological, evolutionary & environmental sciences

For a reference copy of the document with all sections, see nature.com/documents/nr-reporting-summary-flat.pdf

# Life sciences study design

All studies must disclose on these points even when the disclosure is negative.

| | |
|---|---|
| Sample size | Sample sizes for bacteria injections were kept to n=3 per condition due to clear readout of fungal and bacterial viability and colonization and large effect sizes. For adaptive laboratory evolution, sample size of 10 lines was chosen because multiple lines increase the chance for succesful adaptation and increase the chance for allowing comparisons of affected areas in the genome, even though any single line can yield meaningful results by itself. As adaptive laboratory evolution is labor intensive, a trade-off between maximised results and practicality was made at 10 lines. |
| Data exclusions | No data was excluded in the analyses that fulfilled the set standards and thresholds. For bacterial injection into the fungus, injections not resulting in a viable fungal germling with visible intracellular bacteria were excluded. |
| Replication | Successfull bacterial injections were performed in at least triplicates. The adaptive laboratory evolution experiment followed dynamics observed during a pre-trial and was performed in ten lines which showed generally similar trends but also variability in effect sizes. The adaptations by the organisms may not be identically replicated due to the inherent stochasticity in evolutionary experiments. |
| Randomization | Samples were not randomized as the aim of the study was to describe and characterize effects which could not be foreseen. |
| Blinding | Investigators were not blinded due to the study following adaptations over time where negative controls are not applicable. Analysis of collected data between samples followed strictly objective criteria. |

# Reporting for specific materials, systems and methods

We require information from authors about some types of materials, experimental systems and methods used in many studies. Here, indicate whether each material, system or method listed is relevant to your study. If you are not sure if a list item applies to your research, read the appropriate section before selecting a response.

### Materials & experimental systems

| n/a | Involved in the study |
|---|---|
| ☒ | ☐ Antibodies |
| ☒ | ☐ Eukaryotic cell lines |
| ☒ | ☐ Palaeontology and archaeology |
| ☒ | ☐ Animals and other organisms |
| ☒ | ☐ Clinical data |
| ☒ | ☐ Dual use research of concern |
| ☒ | ☐ Plants |

### Methods

| n/a | Involved in the study |
|---|---|
| ☒ | ☐ ChIP-seq |
| ☐ | ☒ Flow cytometry |
| ☒ | ☐ MRI-based neuroimaging |

# Plants

| | |
|---|---|
| Seed stocks | N/A |
| Novel plant genotypes | N/A |
| Authentication | N/A |

# Flow Cytometry

## Plots

Confirm that:

☒ The axis labels state the marker and fluorochrome used (e.g. CD4-FITC).

☒ The axis scales are clearly visible. Include numbers along axes only for bottom left plot of group (a 'group' is an analysis of identical markers).

☒ All plots are contour plots with outliers or pseudocolor plots.

☒ A numerical value for number of cells or percentage (with statistics) is provided.

## Methodology

| | |
|---|---|
| Sample preparation | Spores were harvested 6±1 days after injection or plating from spores. Spore solution (8.5% NaCl, 1% Tween20) was added to plates (16 mL for square plates, 12 mL for round plates), and the spores thoroughly detached using a spatula. Remaining mycelium was clumped up and gently pressed with the spatula to release the spore solution from the mycelium. The spore solution was filtered through a 10 µm CellTrics filter (Sysmex, Germany). Spores were washed three times with 1 mL Hepes2 (8000 rcf, 2 min.) and stored at 4 °C ON for FACS. |
| Instrument | FACSAria Fusion BSL2 Cell sorter (BD, US). |
| Software | FlowJo v10 (BD, US) |
| Cell population abundance | During determination of germination rate of positive spores, per sample 5 germlings were checked under the microscope to be populated with GFP-labeled bacteria. If one germling was not positive, another 5 germlings were checked. No sample had to be dismisssed due to more than 2/10 not being colonized by labeled bacteria. The abundance of positive spores varied greatly throughout the experiments, as it was one of the main features studied. |
| Gating strategy | Single spores were selected using SSC-A/FSC-A and FSC-H/FSC-A gates. Colonization by bacteria was checked using an SSC-A/ eGFP-A gate. A gating strategy figure is included in the SI. The whole dataset is available on Zenodo under following link: 10.5281/zenodo.12189101 |

☒ Tick this box to confirm that a figure exemplifying the gating strategy is provided in the Supplementary Information.

