## [Peer Review File · Nature]

Manuscript Title: Inducing Novel Endosymbioses by Bacteria Implantation in Fungi

Redactions – unpublished data

Reviewer Comments & Author Rebuttals

Reviewer Reports on the Initial Version:

Referees' comments:

Referee #1 (Remarks to the Author):

Giger et al. report an interesting approach to the early stage of the evolution of endosymbiosis. By injecting labelled bacterial cells into fungal hyphae using FluidFM technique, the authors successfully monitored the destiny of “experimental intracellular symbiotic and non-symbiotic bacterial cells” in the fungal cytoplasm at the single-cell level during hyphal growth and spore formation. Furthermore, using FACS-mediated selection of bacteria-transmitting fungal spores, the authors successfully forced the bacteria to evolve symbiotic traits in the non-symbiotic host background.

The model organisms are:

Filamentous fungus *Rhizopus microspores*, naturally symbiotic strain EH and non-symbiotic strain NH
Mycetohabitans rhizoxinica, an endosymbiotic bacterium of *R. microspores*
Escherichia coli, a laboratory bacterium with no relationship to *R. microspores*

I regard this study as an interesting, original, and nicely conducted body of work. The adoption of FluidFM for experimentally establishing bacterial injection into fungal hyphal cells is technically new and impressive. The visualization of bacteria at early stages of infection/proliferation/migration at the single cell level provides important information as to how an initial stage of endosymbiotic association can set in. The positive selection of spore-mediated vertical transmission using FACS works nicely, thereby practically enabling the laboratory experimental evolution between the endocellular bacterium and the fungal host. The observation of hyphal septum formation for confining non-symbiotic bacteria (= *E. coli*) in restricted hyphal regions illustrates a fungal immune mechanism against endocellular intruders. While *E. coli* did not develop symbiotic traits, *M. rhizoxinica* developed symbiotic traits through infection and vertical transmission passages in the non-symbiotic host strain *R. microspores* NH, which can be regarded as representing an early stage of experimental evolution toward endosymbiosis. Development of not only ordinary symbiotic traits (=improved transmission efficiency, mitigated infection cost, etc.) but also potentially beneficial traits (= increased production of rhizoxins) is notable in the context of the evolution of mutualism and evolutionary novelty. Experimental evaluation of fitness costs associated with spore-associated vertical transmission comprises important data. Demonstration of putative evolutionary changes in both host and bacterial sides by transfection experiments with fitness evaluation is fine.

On the other hand, I have an impression that the analyses in this study remain at phenotypic levels. While the co-evolution/co-adaptation of both partners, *R. microspores* NH and *M. rhizoxinica*, was suggested only by transfection experiments with fitness evaluation, genetic/molecular mechanisms underpinning the symbiotic traits are totally untouched. Considering that *M. rhizoxinica* was

originally indispensable endosymbiotic bacteria of *R. microspores* EH, the evolution/adaptation in the conspecific host strain *R. microspores* NH is interesting but not surprising. If *E. coli* could develop symbiotic traits in *R. microspores* as observed in an insect system (ex. Koga et al. 2022 *Nat Microbiol*), it would have been more exciting.

The authors cite only a mini-review (Meaney et al. 2020 *Curr Opin Syst Biol*) and fail to refer to previous important studies on artificial generation of endosymbiosis in laboratory (ex. Mehta et al. 2018 *PNAS*; Su et al. 2022 *Curr Biol*).

The authors state that “To our knowledge, there are no known examples of bacteria using host microtubules for transportation... (L. 117-118)”. To my knowledge, microtubule-mediated transportation of endosymbiotic bacterial within host cells has been reported from *Wolbachia* endosymbionts of insects and other invertebrates (ex. Ferree et al. 2005 *PLoS Pathog*; Serbus et al. 2008 *Annu Rev Genet*).

Referee #2 (Remarks to the Author):

a) Summary of key results. This paper develops a means of transinfection of a symbiont from a fungal strain *Rhizopus* in which it naturally resides, into a closely related strain that is naturally uninfected, and then tracks its evolution in terms of transmission, impact on host and titre, over generations of passage. The symbiont becomes better adapted to the host in some key fitness-related phenotypes, and the host to the microbe as well.

b) Originality and Significance. This paper was not particularly original/significant in my opinion in terms of conceptual advance. It sets up a 'big picture' of a novel approach to examining the evolution of symbiosis through introducing symbionts to novel hosts (transplantation), but fails to really comprehend that this approach has been deployed many times before in studies of insect symbiosis, where the approach is routine, and has been very successful in elucidating changes that occur following host switching. In insect endosymbiosis, the technique is termed transinfection, and is widely used for understanding determinants host range, how symbioses destabilize in novel hosts, and how they evolve in novel hosts.

The first case of transinfection of an insect endosymbiont dates to Maloglowkin & Poulson (1957) <https://www.science.org/doi/10.1126/science.126.3262.32>

This was followed by moving the symbiont to a novel host species in 1965 by Ikeda doi: [10.1126/science.147.3662.1147](https://www.science.org/doi/10.1126/science.147.3662.1147).

Since this time, there have been a number of studies where insect symbionts have been moved from one host species to another; in some of these phenotypes are examined, in others evolution of fitness related phenotypes following passage is tracked.

e.g. McGraw et al 2002 <https://www.pnas.org/doi/full/10.1073/pnas.052466499>

e.g. Tinsley & Majerus 2007 <https://bmcecolevol.biomedcentral.com/articles/10.1186/1471-2148-7-238>

e.g. Nakayama et al 2014 <https://www.nature.com/articles/hdy2014112>

Indeed, the field has some sophisticated studies where engineered environmental strains are introduced to novel hosts to perform symbiotic functions. e.g. Su et al 2022 <https://www.sciencedirect.com/science/article/pii/S0960982222011630>

and studies where even obligate symbionts are transferred and competed: Perreau et al 2021 doi: 10.1073/pnas.2102467118

Further, in some of these studies the evolution is examined directly through genomic changes over passage such that genetic evolution is directly measured:

Woolfit et al 2012 <https://doi.org/10.1093/gbe/evt169>

My conclusion thus is that the conceptual findings are not particular original/significant - evolution to lower virulence on passage is well known, adaptive evolution on early host switching well known, and the basic technique of introducing a symbiont to a novel host very well established as a basis for advancing understanding.

c) Data and methodology

The methodology is very sophisticated in places.

The main area where work should go further is in cementing evolutionary change at the genomic level. Symbiont-host interactions can vary due to titre, or even gene expression of host and symbiont, and it is important to solidify the degree of genetic change through comparative genomics of evolved microbes and their hosts (where host evolution is also concluded, as here).

I have a little scepticism about the host evolution, as the starting material is a 'strain' which one presumes has no standing genetic variation, and evolution usually is slow in small populations without abundant genetic variation - so I think making this result robust through sequencing is important (note small here is related to mutation rate; bacterial experimental evolution works as population sizes are of the order of 10 billion or more, such that with bacterial mutation rates, all mutations happen each division in this population size just about; the population sizes of the fungus here is quite a bit lower and the mutation rate lower, hence the scepticism about it evolving).

d) Statistics and uncertainties. The selection experiments have an odd design - 10 lines that are reduced to 3 + 1 pool, and the 'no selection' design with I think just three replicates. I am not clear how the first rather odd design came about, and whether 3 replicates is satisfactory for the latter (which I think only derived from one of the adapted strains, but may be wrong in this).

e) Robustness and validity. Generally very high for the actual experiment due to the FACS sorting providing large sample sizes.

However, the selection experiment is oddly designed and the lack of genome sequencing leaves us dependent on phenotype for understanding evolution in a system where load and gene expression may also have a heritability.

I think one might argue also that more general messages would be obtained if transplantation was done into more than one *Rhizopus* strain.

f) Improvements: Genomic validation of adaptive change in host and microbe is essential and relatively simple to achieve; replication to more than a single host line would be desirable.

g) References to previous work. As argued above, this work is not as novel as the account suggests - an entire literature on this topic has, in my opinion, been completely sidelined.

h) Clarity and context. It is a nicely written piece but does ignore an awful lot of previous transinfection work such that this manuscript sells this approach as novel when it is in my opinion only novel in technique of transinfection and the cleverness of the FACS design for handling the experiment.

Referee #3 (Remarks to the Author):

The work submitted by Giger, Ernst, Richter, et al. investigates the implantation of bacteria into fungi as a novel approach to establishing and studying endosymbioses between bacteria and fungi, which is exciting and relevant for fields such as fungal biology and evolution, as well as environmental and clinical mycology.

The authors used two single-cell techniques, namely fluidic force microscopy (FluidFM) and fluorescent activated cell sorting (FACS), to implant bacteria in fungal germlins and monitor bacterial transmission within the asexual spores, respectively. An essential aspect of their experimental design was using *Rhizopus microsporus* as the fungal model. This fungal species is coenocytic/aseptate and has naturally occurring strains that establish symbioses with Mycetohabitans bacteria (authors called them EH for endosymbiont-harboring), and others that do not (non-harboring, NH). This fact allowed authors to compare the implantation of the non-symbiotic bacterium *Escherichia coli* with the effects of the endosymbiotic *M. rhizoxinica* in both EH and NH strains of *R. microsporus*.

First, the authors validated their procedures by injecting *M. rhizoxinica* (GFP-labeled) into its natural host, *R. microsporus* EH, reconstituting the natural symbiosis and its vertical transmission. Then, they observed the phenotypes produced by the implantation of GFP-labeled *E. coli* within *R. microsporus* EH and NH. These observations revealed that both fungus and bacteria survived implantation, that *E. coli* cells proliferated rapidly, and that the fungus reacted with septa formation, separating cells filled with bacteria and without them. They had beautiful videos showing *E. coli*-filled compartments bursting out, and the separation of bacteria-filled and bacteria-free compartments was visible. Also, they confirmed that although some *E. coli* cells survive in low densities in the mycelium, the bacteria were not inherited in the spores.

When similar experiments were done with *M. rhizoxinica* in the NH strain of *R. microsporus*, authors observed that these bacteria survived and moved similarly as they did in the EH strain (likely using microtubules), and that although spore formation is independent of bacteria, some of the spores did contain bacterial cells (less than 4%).

They investigated the stability of the novel and synthetic symbiosis along ten generation rounds, always selecting the B+ spores. They evaluated the capacity of the resulting B+ and B- spores to

germinate and the abundance of B+ spores. The authors made a fitness index (germination*abundance) from these two measurements and also measured the production of rhizoxin congeners by synthetic fungal holobionts. Their results and analyses showed that germination success increases consistently along generations, being no more difference between B+ and B- spores at round 10. Also, the proportion of spores containing Mycetohabitans bacteria increased significantly from round 3 to 7. However, B+ spores showed delayed germination compared to B- spores. Notably, the ten generated symbiotic lines produced rhizoxin and some of its congeners.

In order to evaluate if one or the two partners adapted to the evolution experiment, they performed cross-infection experiments with the ancestral and evolved bacteria and fungi. These experiments revealed that both partners adapted during the experiment. Noteworthy, the evolved fungus performed better with the ancestral bacteria regarding germination (%). The authors also estimated the load of bacteria in the B+ population to reveal that the ancestral synthetic holobiont had, in general, a higher bacterial load than the evolved population. Germination success was not, however, significantly correlated with bacterial load. Remarkably, bacterial presence in the spores in the ancestral and evolved symbioses reduced its germination as spores aged (27 days).

Finally, the authors tested by wet-lab experiments and by employing a mathematical model, what happens at the population level if no selection of the B+ spores is applied. They could show that the forced symbiosis is lost more rapidly (1 round) in the ancestral pair than in the evolved partnership (4-5 rounds).

Altogether, the results presented are novel and solid and shed light, quantitatively, on the stability, fitness, and costs of this new fungal-bacterial symbiosis. I enjoyed reading the manuscript (and watching the videos!) and believe this approach will help answer many other open and intriguing questions, such as which molecular responses were in place in both partners to enable adaptation. Did all lines follow the same trajectories? How many generations are needed to achieve 100% vertical transmission? Would the use of *R. delemar* or other *Rhizopus* species render similar results? Etc.

Some other comments to improve the manuscript and the presentation of the results are the following:

1. The term "germination rate" seems unsuitable. Instead, the terms germination (%) or germination success (%) better reflect what the authors measured. How many spores germinated from the total (either for the B+ or B- populations), right? Germination rate would express the number of spores that germinated per hour or per unit of time (a rate). The authors used both in the text and figures, as they would be the same. See, for example, Figs. 4a, 5a and 5c. Please check consistency throughout the manuscript, figures, and supplemental materials.
2. The term "abundance (%)" is also less descriptive and precise than "% vertical transmission". The authors refer to the proportion of spores in the new generation that were sorted and confirmed as B+ from the total. Abundance can be confounded with bacterial load within a spore.
3. How was the mathematical model validated? Can authors provide more details on this?

Some other minor comments are:

4. Legends could be more explicit also if the B+ or B- were used instead of just "Positive" or "Negative" in Fig. 4.
5. In Fig. 5a, labels are inconsistent with the text (line 208, page 5). Probably Fanc/Bevo is better than the other way around.
6. Legend of Fig. 3. The signal is absent "after injection with *E. coli* and in the non-injected negative

control." for consistency with the figure.

Abstract:

Line 34. It is advisable to mention which selection was imposed. Thus, I suggest changing "Positive selection improved..." to "Strict selection of spores harboring bacterial symbionts improved their transmission over generations..."

Introduction:

Line 40. "Intracellular endosymbioses" sound tautologic. Probably just "Intracellular symbioses" or "Endosymbioses are examples..."

Results:

Line 100: used "attempted" instead of "had not been achieved," as placement of Mycetohabitans in EH strains has been achieved by Micro-injection employing a laser beam as reported previously in Partida-Martinez et al. Curr Biol. 2007 (Ref 20).

Line 144: What units apply to (detection limit 10^{-6})?

Methods:

Line 540: Please specify how germlins were checked to confirm fluorescent bacteria's presence. Was it microscopically? By PCR?..

Line 582: Was solvent A 0.1% (v/v) formic acid in which solvent? Please specify.

Author Rebuttals to Initial Comments:

Referee expertise:

Referee #1: endosymbiosis, evolution

Referee #2: evolution and ecology of symbiosis, genomics, mathematical modelling

Referee #3: bacterial endosymbionts of fungi

Referees' comments:

Referee #1 (Remarks to the Author):

Giger et al. report an interesting approach to the early stage of the evolution of endosymbiosis. By injecting labelled bacterial cells into fungal hyphae using FluidFM technique, the authors successfully monitored the destiny of “experimental intracellular symbiotic and non-symbiotic bacterial cells” in the fungal cytoplasm at the single-cell level during hyphal growth and spore formation. Furthermore, using FACS-mediated selection of bacteria-transmitting fungal spores, the authors successfully forced the bacteria to evolve symbiotic traits in the non-symbiotic host background.

The model organisms are:

Filamentous fungus *Rhizopus microspores*, naturally symbiotic strain EH and non-symbiotic strain NH

Mycetohabitans rhizoxinica, an endosymbiotic bacterium of *R. microspores*

Escherichia coli, a laboratory bacterium with no relationship to *R. microspores*

I regard this study as an interesting, original, and nicely conducted body of work. The adoption of FluidFM for experimentally establishing bacterial injection into fungal hyphal cells is technically new and impressive. The visualization of bacteria at early stages of infection/proliferation/migration at the single cell level provides important information as to how an initial stage of endosymbiotic association can set in. The positive selection of spore-mediated vertical transmission using FACS works nicely, thereby practically enabling the laboratory experimental evolution between the endocellular bacterium and the fungal host. The observation of hyphal septum formation for confining non-symbiotic bacteria (= *E. coli*) in restricted hyphal regions illustrates a fungal immune mechanism against endocellular intruders. While *E. coli* did not develop symbiotic traits, *M. rhizoxinica* developed symbiotic traits through infection and vertical transmission passages in the non-symbiotic host strain *R. microspores* NH, which can be regarded as representing an early stage of experimental evolution toward endosymbiosis. Development of not only ordinary symbiotic traits (=improved transmission efficiency, mitigated infection cost, etc.) but also potentially beneficial traits (= increased production of rhizoxins) is notable in the context of the evolution of mutualism and evolutionary novelty. Experimental evaluation of fitness costs associated with spore-associated vertical transmission comprises important data. Demonstration of putative evolutionary changes in both host and bacterial sides by transfection experiments with fitness evaluation is fine.

Thank you for the positive assessment and for highlighting the novelty in terms of the technology and the mutualistic and evolutionary outcome.

On the other hand, I have an impression that the analyses in this study remain at phenotypic levels. While the co-evolution/co-adaptation of both partners, *R. microspores* NH and *M. rhizoxinica*, was suggested only by transfection experiments with fitness evaluation, genetic/molecular mechanisms underpinning the symbiotic traits are totally untouched. Considering that *M. rhizoxinica* was originally

indispensable endosymbiotic bacteria of *R. microspores* EH, the evolution/adaptation in the conspecific host strain *R. microspores* NH is interesting but not surprising. If *E. coli* could develop symbiotic traits in *R. microspores* as observed in an insect system (ex. Koga et al. 2022 Nat Microbiol), it would have been more exciting.

To extend our fitness evaluation to the genetic level, we now sequenced the fungus and endosymbiont throughout the passages for the line that we had phenotypically characterized throughout ('line 4') using long and short read sequencing. In addition, we sequenced all parallel lines at passage 7 and 10. For the evolutionary trajectory, we show that genetic changes occurred in the fungal population during the passaging, but not in the bacterial population. We also show that the frequency of mutant alleles in the fungus correlates with fitness improvement (Figure 5a). This finding confirms our cross-injection validation experiment (F_{Evo}/B_{Anc}), which had already indicated phenotypically that the fungus had evolved. Sequencing of the fungus from the reinjection experiment (F_{Evo}/B_{Anc}) confirmed the presence of the mutations in the validation experiment. The enriched mutations suggest changes related to transcription/translation, endocytosis, mitochondrial maintenance, and ion transport (Suppl. Fig. 4 and Suppl. Table 1).

The reviewer mentions that the successful adaptive evolution *M. rhizoxinica*-*R. microsporus* NH is not surprising. However, we would like to point out that the induction of novel intracellular endosymbioses in fungi is extremely rare, technically challenging to achieve experimentally, and has not been reported in a prospective, highly resolved manner. We have engineered a cytosolic endosymbiosis in a filamentous fungus and show both phenotypic and genomic adaptation. Strikingly, even in the non-host pairing, for which we expected a low but still possible successful adaptive evolution, the initial fitness index of the endosymbiosis outcome was only 0.000006 (requiring a high sensitivity FACS assay for selection) and improved under strict selection >40'000 fold. We find this impressive adaption of the system surprising because it highlights plasticity. We find it even more surprising that the fungus evolved rather than the bacterium. Such host adaptation is not seen in insect systems.

[REDACTED]

Regarding the work of Koga *et al.* 2022 Nat Microbiol, we agree that this is an interesting paper dealing with the evolution of a symbiotic relationship. However, their finding of establishing *E. coli* as an extracellular colonizer of a specialized gut compartment in a metabolically dependent insect host cannot be directly compared to establishing *E. coli* as an intracellular endosymbiont in a filamentous fungus. In insects, endosymbiont replacement is observed regularly (McCutcheon *et al.* 2019 Curr Biol) and is often facilitated by evolutionary adaptations and a metabolic dependence of the host. Furthermore, in the system of Koga *et al.*, *E. coli* belongs to the same family of Enterobacteriaceae as the original *Pantoea* symbiont.

In conclusion, we share the reviewer's considerations on where higher compatibility might be expected and believe that the novelty of our study lies in the successful induction endosymbiosis in an endosymbiont-independent host and the demonstration of stabilization of the interaction through selection. We believe that the successful expansion of the host range of a bacterium to a soil-dwelling fungus lays the groundwork for future work on even more remote partnerships. We have rephrased the manuscript in several places to better highlight the novelty of our study.

[REDACTED]

The authors cite only a mini-review (Meaney et al. 2020 Curr Opin Syst Biol) and fail to refer to previous important studies on artificial generation of endosymbiosis in laboratory (ex. Mehta et al. 2018 PNAS; Su et al. 2022 Curr Biol).

We have adapted the manuscript to include the studies by Mehta *et al.* and Su *et al.*

Regarding the study by Mehta *et al.*, the authors aimed to embed *E. coli* within budding yeast, but left pivotal questions unanswered, notably the demonstration of vertical inheritance and the perpetuation of the symbiotic relationships, because co-cultures were re-streaked on agar plates and no single clones propagated. In our view, the experiments may not fully support the claims made in the paper. Briefly, the authors try to show that they can select for chimeras under non-fermentable conditions where the host cell depends on the *E. coli* to supply all ATP. However, even their most stringent selection medium contains 3% of rich cell extracts (1% yeast extract, 2% bacto peptone). This medium is not well defined to allow clear conclusions on the metabolic pathways used for growth. Furthermore, the authors do not document the intracellular localization of the bacterial cells well. Despite the bacteria being labelled with GFP, they do not show confocal microscopy but instead use soft X-ray tomography, which is quite unspecific, and TIRF imaging, which has extremely shallow depth of about 100 nm, as well as a probe which is supposed to stain bacterial rRNA. They also use qPCR to assess the ratio of bacterial genomes to yeast genomes and find even under “strict” selection a ratio of down to 0.083, which poses the question where “the other yeast cells” (the majority) get their ATP from. The authors claim that per re-plating event, the cells undergo 40 doublings but do not provide data to support this, and state that the first round of plating and sometimes the second round of plating is done on selection medium II which contains 0.1% fermentable glucose (in addition to 3% rich compounds). They mention a single *E. coli* which could be isolated after 120 doublings. Thus, the authors’ conclusion on a vertical transmission seems not supported by the data and no video material of vertical transmission is provided. The authors themselves state the limitation in a follow up paper (Cournoyer *et al.* 2022, Nature Comm.). However, we agree it should be referred to in our study. To keep it succinct, we now refer to this paper as follows: “Mehta *et al.* attempted to install an engineered *E. coli* to supply the yeast *Saccharomyces cerevisiae* host cell with ATP, and Cournoyer *et al.* aimed to install cyanobacteria in the same host as a photosynthetic organelle, but these experiments did not result in endosymbioses that could be stably propagated under strict selection.”

Regarding Su *et al.*: This is indeed an interesting study on an insect system, for which it is well documented that replacements of endosymbionts occurred throughout evolution repeatedly. We now refer to this paper and provide more context on insect endosymbiosis in the introduction and the discussion.

The authors state that “To our knowledge, there are no known examples of bacteria using host microtubules for transportation... (L. 117-118)”. To my knowledge, microtubule-mediated transportation of endosymbiotic bacterial within host cells has been reported from Wolbachia endosymbionts of insects and other invertebrates (ex. Ferree et al. 2005 PLoS Pathog ; Serbus et al. 2008 Annu Rev Genet).

We thank the referee for making us aware of the studies on Wolbachia localization in *Drosophila* oocytes. We have adapted the manuscript accordingly and now include the information on microtubule mediated transport of Wolbachia-containing vesicles.

We believe, however, that the microtubule-based transport of a cytosolic bacterium (not enclosed in a vesicle) is novel and intriguing with respect to the filamentous morphology of the fungal host. The bacteria must be able to move within the coenocytic mycelium to reach the sites of spore formation to allow vertical transmission.

To further strengthen our conclusion on microtubule-based transport, we have now conducted an additional experiment and added the data showing the effect of dynein inhibition by ciliobrevin D on the intracellular movement of *M. rhizoxinica* (Suppl. Video 4).

Referee #2 (Remarks to the Author):

a) Summary of key results. This paper develops a means of transinfection of a symbiont from a fungal strain *Rhizopus* in which it naturally resides, into a closely related strain that is naturally uninfected, and then tracks its evolution in terms of transmission, impact on host and titre, over generations of passage. The symbiont becomes better adapted to the host in some key fitness-related phenotypes, and the host to the microbe as well.

b) Originality and Significance. This paper was not particularly original/significant in my opinion in terms of conceptual advance. It sets up a ‘big picture’ of a novel approach to examining the evolution of symbiosis through introducing symbionts to novel hosts (transplantation), but fails to really comprehend that this approach has been deployed many times before in studies of insect symbiosis, where the approach is routine, and has been very successful in elucidating changes that occur following host switching. In insect endosymbiosis, the technique is termed transinfection, and is widely used for understanding determinants host range, how symbioses destabilize in novel hosts, and how they evolve in novel hosts.

The first case of transinfection of an Insect endosymbiont dates to Maloglowkin & Poulson (1957) <https://www.science.org/doi/10.1126/science.126.3262.32>

This was followed by moving the symbiont to a novel host species in 1965 by Ikeda doi: [10.1126/science.147.3662.1147](https://www.science.org/doi/10.1126/science.147.3662.1147).

Since this time, there have been a number of studies where insect symbionts have been moved from one host species to another; in some of these phenotypes are examined, in others evolution of fitness related phenotypes following passage is tracked.

e.g. McGraw et al 2002 <https://www.pnas.org/doi/full/10.1073/pnas.052466499>

e.g. Tinsley & Majerus 2007 <https://bmcecol.evol.biomedcentral.com/articles/10.1186/1471-2148-7-238>

e.g. Nakayama et al 2014 <https://www.nature.com/articles/hdy2014112>

Indeed, the field has some sophisticated studies where engineered environmental strains are introduced to novel hosts to perform symbiotic functions. e.g. Su et al 2022 <https://www.sciencedirect.com/science/article/pii/S0960982222011630>

and studies where even obligate symbionts are transferred and competed: Perreau et al 2021 doi: 10.1073/pnas.2102467118

Further, in some of these studies the evolution is examined directly through genomic changes over passage such that genetic evolution is directly measured:

Woolfit et al 2012 <https://doi.org/10.1093/gbe/evt169>

My conclusion thus is that the conceptual findings are not particular original/significant– evolution to lower virulence on passage is well known, adaptive evolution on early host switching well known, and the basic technique of introducing a symbiont to a novel host very well established as a basis for advancing understanding.

We believe that our work offers substantial novelty to published research on endosymbiogenesis and has significant novelty compared to established findings in insect endosymbionts. Our work revolves around a host with an evolutionarily more ancient lifeform, an aseptate filamentous fungus. In this system, the endosymbiont resides in the cytosol (without a host membrane around it) and distributes throughout the whole organism. This stands in contrast to insect endosymbionts, where the complex host has evolved specific tissues in which the endosymbionts are housed, oftentimes extracellularly or within specialized organelles. Moreover, insects often rely on metabolic supplementation by their endosymbionts. These factors all play a role in the abundance of natural endosymbiont replacements, and the relative ease of interventional transinfection. However, these specializations also make it difficult to generalize findings in insects to other phyla, where the lack of similar specializations may explain why endosymbiont transplantation has remained so elusive. The knowledge about the dynamics in transinfection in insects does not preclude novelty when endosymbiosis is induced and described in other phyla.

By inducing an endosymbiosis between bacteria and fungi, we contribute to the expansion of experimental endosymbiogenesis to a fast-growing and relevant field of research. Bacterial-fungal endosymbioses are of high ecological importance, from well-known examples such as plant-beneficial arbuscular mycorrhizal fungi to an ever-increasing number of symbionts detected in a wide range of hosts (e.g. Bonfante *et al.* ISME J 2017). Yet, methods for endosymbiont transplantation have been very limited. Through our approach with the FluidFM and live observation of the bacterial injection, followed by long and stable propagation, extensive phenotypic observations, and now also genomic characterization, we can follow the induced endosymbiosis in unprecedented resolution and on complementary levels. We show that some dynamics resemble observations made in insects, such as attenuation of bacterial pathogenicity, while also uncovering new dynamics, such as genomic adaptation of the host organism. And we believe that the cytosolic placement of the bacterium can help to find more broadly applicable methods for stable coexistence which in the future may be more easily transferred to different pairings. Nevertheless, we have adapted the manuscript to include more literature on the description of transinfection in insects in the introduction and to provide more context of our findings to observations made in insects in the discussion.

Regarding the addressed literature:

(Maloglowkin & Poulson, Ikeda, Tinsley & Majerus, and Nakayama et al.) Spiroplasma are fascinating bacterial endosymbionts with intricate host manipulating mechanisms. They have specialized in infecting a multitude of hosts and fast changes to its phenotypic effects on hosts have been described. We believe that our experiments show a clear difference to work done with spiroplasma due to the choice of host, choice of endosymbiont, and the direct cytosolic as opposed to vacuolar localization of the bacterium. It is reassuring that observations made in our system reflect some adaptations that are expected based on established research on insect endosymbiosis such as pathogenicity attenuation in a new host. We have added a reference to Nakayama *et al.*

We have also added a reference to McGraw *et al.* Based on our genetic insights, we believe that bacterial density in spores may be controlled more by the host rather than the bacterial endosymbiont.

The work of Su *et al.* is interesting, and the engineering of the endosymbiont to reduce the negative impact on host amino acids is of relevance for future applications of rational mutualistic endosymbiont design. We have adapted our manuscript to include the reference to Su *et al.* Our work differs from Su *et al.* in the use of a different host organism which does not have evolved bacteriomes to house bacterial endosymbionts, and in using a prototrophic rather than an auxotrophic host. We also demonstrate the transplantation of a new-to-host metabolic function with the endosymbiont.

Perreau *et al.* investigate within-host selection in aphids which is of importance for the effectiveness of natural selection on obligate endosymbionts and to understand the different effects of within-host selection, between-host selection, and random fixation of mutations through bottleneck events by creating heteroplasmic aphids. We believe the system is fundamentally different from the *Rhizopus* system, due to the mixing of nuclei in the aseptate fungus and the asexual reproduction in the absence of a dedicated germline.

Woolfit *et al.* is another example of informative research in insects that we have added to our manuscript. Again, the system is profoundly different from the one we studied. Also, in contrast to this study, we could not detect any SNPs taking over the population for the bacterium but observed changes in the host's genome.

c) Data and methodology

The methodology is very sophisticated in places.

The main area where work should go further is in cementing evolutionary change at the genomic level. Symbiont-host interactions can vary due to titre, or even gene expression of host and symbiont, and it is important to solidify the degree of genetic change through comparative genomics of evolved microbes and their hosts (where host evolution is also concluded, as here).

We have added an analysis of the evolutionary change at the genomic level of the host. This demonstrates that there are fundamental differences between our system and insect endosymbionts.

I have a little scepticism about the host evolution, as the starting material is a 'strain' which one presumes has no standing genetic variation, and evolution usually is slow in small populations without abundant genetic variation - so I think making this result robust through sequencing is important (note small here is related to mutation rate; bacterial experimental evolution works as population sizes are of the order of 10 billion or more, such that with bacterial mutation rates, all mutations happen each division in this population size just about; the population sizes of the fungus here is quite a bit lower and the mutation rate lower, hence the scepticism about it evolving).

We would like to caution the conclusion on lack of genetic variation in the fungal host. The fungus spores contain several nuclei and they undergo a large number of divisions in the resulting coenocytic mycelium, the nuclei divide individually and can incur mutations during every duplication event. During coenocytic growth, phenotypes of mutations may be partially masked by other nuclei. Packaging of several nuclei into sporangiospores leads to new evolutionary units and increases the likelihood for evolutionary units carrying specific mutations. While the experiment starts with injection into an organism with few nuclei, variation is quickly generated throughout the experiment. Our sequencing data support the high variation and in consequence genomic plasticity of offspring.

d) Statistics and uncertainties. The selection experiments have an odd design - 10 lines that are reduced to 3 + 1 pool, and the 'no selection' design with I think just three replicates. I am not clear how the first rather odd design came about, and whether 3 replicates is satisfactory for the latter (which I think only derived from one of the adapted strains, but may be wrong in this).

To provide a better overview on the experimental design, we have now moved the overview of the adaptive evolution experiment to the main figure (Fig. 3a). The reduction to 3 lines plus pool of all remaining lines after round 7 was due to experimental limitations with the available resources at the FACS core facility. However, it allowed us to focus on the best performing lines, which were of particular interest. Based on the variability observed in the "no selection" design, the 3 replicates are significant to show the difference between the two conditions.

e) Robustness and validity. Generally very high for the actual experiment due to the FACS sorting providing large sample sizes.

The data points obtained by FACS sorting are indeed high (100'000-1'000'000 data points per sample). Our data reflect variation that is inherent to the biology.

However, the selection experiment is oddly designed and the lack of genome sequencing leaves us dependent on phenotype for understanding evolution in a system where load and gene expression may also have a heritability.

As mentioned above, we now conducted genome sequencing. We think we now adequately addressed the concern with our new data.

I think one might argue also that more general messages would be obtained if transplantation was done into more than one *Rhizopus* strain.

[REDACTED]

f) Improvements: Genomic validation of adaptive change in host and microbe is essential and relatively simple to achieve; replication to more than a single host line would be desirable.

Genomic validation has been added to the manuscript. Replication of the evolution experiments is out of scope in our view.

g) References to previous work. As argued above, this work is not as novel as the account suggests - an entire literature on this topic has, in my opinion, been completely sidelined.

We have focused our work on fungi, as the title indicated. We have clarified this now better in the main body of the manuscript and added references to other work performed on this topic at several places, i.e. introduction, results, discussion.

h) Clarity and context. It is a nicely written piece but does ignore an awful lot of previous transfection work such that this manuscript sells this approach as novel when it is in my opinion only novel in technique of transfection and the cleverness of the FACS design for handling the experiment.

Transfections are not commonly observed in fungi in contrast to insects, and the systems are fundamentally different regarding their biology. We have adapted the manuscript to include more references to work performed on transfection in insects.

Referee #3 (Remarks to the Author):

The work submitted by Giger, Ernst, Richter, et al. investigates the implantation of bacteria into fungi as a novel approach to establishing and studying endosymbioses between bacteria and fungi, which is exciting and relevant for fields such as fungal biology and evolution, as well as environmental and clinical mycology.

The authors used two single-cell techniques, namely fluidic force microscopy (FluidFM) and fluorescent activated cell sorting (FACS), to implant bacteria in fungal germlins and monitor bacterial transmission within the asexual spores, respectively. An essential aspect of their experimental design was using *Rhizopus microsporus* as the fungal model. This fungal species is coenocytic/aseptate and has naturally occurring strains that establish symbioses with Mycetohabitans bacteria (authors called them EH for endosymbiont-harboring), and others that do not (non-harboring, NH). This fact allowed authors to compare the implantation of the non-symbiotic bacterium *Escherichia coli* with the effects of the endosymbiotic *M. rhizoxinica* in both EH and NH strains of *R. microsporus*.

First, the authors validated their procedures by injecting *M. rhizoxinica* (GFP-labeled) into its natural host, *R. microsporus* EH, reconstituting the natural symbiosis and its vertical transmission. Then, they observed the phenotypes produced by the implantation of GFP-labeled *E. coli* within *R. microsporus* EH and NH. These observations revealed that both fungus and bacteria survived implantation, that *E. coli* cells proliferated rapidly, and that the fungus reacted with septa formation, separating cells filled with bacteria and without them. They had beautiful videos showing *E. coli*-filled compartments bursting out, and the separation of bacteria-filled and bacteria-free compartments was visible. Also, they confirmed that although some *E. coli* cells survive in low densities in the mycelium, the bacteria were not inherited in the spores.

When similar experiments were done with *M. rhizoxinica* in the NH strain of *R. microsporus*, authors observed that these bacteria survived and moved similarly as they did in the EH strain (likely using microtubules), and that although spore formation is independent of bacteria, some of the spores did contain bacterial cells (less than 4%).

They investigated the stability of the novel and synthetic symbiosis along ten generation rounds, always selecting the B+ spores. They evaluated the capacity of the resulting B+ and B- spores to germinate and the abundance of B+ spores. The authors made a fitness index (germination*abundance) from these two measurements and also measured the production of rhizoxin congeners by synthetic fungal holobionts. Their results and analyses showed that germination success increases consistently along generations, being no more difference between B+ and B- spores at round 10. Also, the proportion of spores containing Mycetohabitans bacteria increased significantly from round 3 to 7. However, B+

spores showed delayed germination compared to B- spores. Notably, the ten generated symbiotic lines produced rhizoxin and some of its congeners.

In order to evaluate if one or the two partners adapted to the evolution experiment, they performed cross-infection experiments with the ancestral and evolved bacteria and fungi. These experiments revealed that both partners adapted during the experiment. Noteworthy, the evolved fungus performed better with the ancestral bacteria regarding germination (%). The authors also estimated the load of bacteria in the B+ population to reveal that the ancestral synthetic holobiont had, in general, a higher bacterial load than the evolved population. Germination success was not, however, significantly correlated with bacterial load. Remarkably, bacterial presence in the spores in the ancestral and evolved symbioses reduced its germination as spores aged (27 days).

Finally, the authors tested by wet-lab experiments and by employing a mathematical model, what happens at the population level if no selection of the B+ spores is applied. They could show that the forced symbiosis is lost more rapidly (1 round) in the ancestral pair than in the evolved partnership (4-5 rounds).

Altogether, the results presented are novel and solid and shed light, quantitatively, on the stability, fitness, and costs of this new fungal-bacterial symbiosis. I enjoyed reading the manuscript (and watching the videos!) and believe this approach will help answer many other open and intriguing questions, such as which molecular responses were in place in both partners to enable adaptation. Did all lines follow the same trajectories? How many generations are needed to achieve 100% vertical transmission? Would the use of *R. delemar* or other *Rhizopus* species render similar results? Etc.

We thank the referee for this positive assessment of our work and the shared anticipation for future work.

Some other comments to improve the manuscript and the presentation of the results are the following:

1. The term "germination rate" seems unsuitable. Instead, the terms germination (%) or gemination success (%) better reflect what the authors measured. How many spores germinated from the total (either for the B+ or B- populations), right? Germination rate would express the number of spores that germinated per hour or per unit of time (a rate). The authors used both in the text and figures, as they would be the same. See, for example, Figs. 4a, 5a and 5c. Please check consistency throughout the manuscript, figures, and supplemental materials.

We have changed the term throughout the manuscript to germination success to prevent confusions (the rate was meant per unit of one, i.e. per round of sporulation).

2. The term "abundance (%)" is also less descriptive and precise than "% vertical transmission". The authors refer to the proportion of spores in the new generation that were sorted and confirmed as B+ from the total. Abundance can be confounded with bacterial load within a spore.

Abundance may indeed be confused with bacterial load, which is not intended. We have now changed the term abundance to "positive fraction". We believe that vertical transmission is related to what we refer to as fitness index, that is, how many of the offspring harbor bacteria. Not all B+ spores germinate and therefore do not contribute to vertical transmission.

3. How was the mathematical model validated? Can authors provide more details on this?

The mathematical model contains our assumptions about the main effects that determine the fate of the B+ population, expressed in numbers. Its validation is provided by the experimental data itself. We rephrased the corresponding section in the Methods to improve clarity.

Some other minor comments are:

4. Legends could be more explicit also if the B+ or B- were used instead of just "Positive" or "Negative" in Fig. 4.

We have changed the figure legends to be more explicit.

5. In Fig. 5a, labels are inconsistent with the text (line 208, page 5). Probably Fanc/Bevo is better than the other way around.

We have changed the figure legends to be consistent with the text.

6. Legend of Fig. 3. The signal is absent "after injection with E. coli and in the non-injected negative control." for consistency with the figure.

We have changed the legend to be more consistent with the figure.

Abstract:

Line 34. It is advisable to mention which selection was imposed. Thus, I suggest changing "Positive selection improved..." to "Strict selection of spores harboring bacterial symbionts improved their transmission over generations..."

We prefer, if possible, to keep "positive selection" as it is commonly used in the context of adaptive evolution experiments. We agree that we should specify selection of spores. We have changed the sentence to: "Positive selection of spores harboring bacterial symbionts improved their transmission over generations, mitigating these fitness constraints and stabilizing the endosymbiosis."

Introduction:

Line 40. "Intracellular endosymbioses" sound tautologic. Probably just "Intracellular symbioses" or "Endosymbioses are examples..."

We agree with the referee that the term is somewhat tautologic, but for the purpose of precision we prefer to keep the wording as is. Endosymbiosis in its broader meaning may also encompass, for example, gut bacteria. And while intracellular does presume "endo" already, we prefer to use endosymbiosis again due to the term's commonness.

Results:

Line 100: used "attempted" instead of "had not been achieved," as placement of Mycetohabitans in EH strains has been achieved by Micro-injection employing a laser beam as reported previously in Partida-Martinez et al. Curr Biol. 2007 (Ref 20).

We have clarified the text to make sure readers understand that we were referring to the FluidFM.

Line 144: What units apply to (detection limit 10⁻⁶)?

The detection limit describes the share of B+ spores. We have changed the section to be more explicit.

Methods:

Line 540: Please specify how germlins were checked to confirm fluorescent bacteria's presence. Was it microscopically? By PCR?..

They were checked microscopically. We have added the information to the manuscript.

Line 582: Was solvent A 0.1% (v/v) formic acid in which solvent? Please specify.

We have adapted the manuscript to specify formic acid in water.

We thank the referee for the helpful improvements.

Reviewer Reports on the First Revision:

Referees' comments:

Referee #1 (Remarks to the Author):

The manuscript has been properly revised. In particular, the addition of genomic data is fine. I have no further comments.

Referee #2 (Remarks to the Author):

a) The piece develops a method for transfer of symbionts between isolates of the same species of fungi and have a strong positive selection to maintain the strain to enable evolution. Ultimately the symbiont-host combination becomes a better fit but is still lost when selection is released. Some genomic data underpinning the changes is now presented.

b) I still regard the piece as of medium high novelty, because symbiont transfer has quite an old history in insects, and because I was not persuaded by a conceptual advance here (I accept a rather elegant technical advance that enables work in fungi). Whilst there are biological distinctions that certainly make this meritorious science, and it was interesting, I think its impact is to broaden the field of enquiry rather than establish new conceptual knowledge. Of course, how important that is will always be subjective.

I'd also note in opposition to the rebuttal - many insect transinfection studies are not conducted on strongly nutritional symbionts and do not involve host's with housing organs. Most are Wolbachia or Spiroplasma which are facultative, generally not housed in bacteriomes, and have weak nutritional impacts. Of these, Wolbachia sits in a vacuole, but I think Spiroplasma sits in the cytosol directly (though can also be extracellular). Of course it nevertheless remains interesting to expand more widely.

Whilst the title is fungal specific, the abstract still makes it appear transinfection/transplantation study is novel to endosymbiosis research: 'However, the emergence of novel endosymbioses is rare and challenging to uncover in retrospect.' is true of fungi - it is not true of endosymbiosis in general.

The last line of the conclusion. 'The production of rhizoxin by strain NH demonstrates that metabolic capabilities can be transferred to new organisms by artificially inducing an endosymbiosis through implantation and subsequent artificial selection.' similarly is not so novel - Wolbachia from *Drosophila melanogaster*, placed into mosquito cell lines and selected, placed into mosquitoes and released to control dengue in the wet tropics is an example of transfer of traits.

c) Data and methodology. The injection and the positive selection through cell sorting are of course novel and technically very elegant.

It was good to see the authors conducting the suggested genomic analysis, but I was not so

persuaded of the results from these. It is clear that the bacterial genome is not responsive during this process. It is argued the fungal genome is responsive. However, the timeline of mutations going to high frequency and phenotype changes for fitness don't match well. For instance, in the most intensely monitored - line 4 - in Figure S1, the fitness index for this line increases between generation 4 and 8, but mutations only appear at decent rate from generation 8 (Figure S4). Line 7 appears to achieve improved fitness without any high frequency mutations in the host occurring. The mutations themselves are not obvious strong candidates either - many intergenic or intronic; not impossible to get a phenotype but not clear. These observations leads me to believe that most of what is going on is likely selection for transgenerational plastic response.

D,E. See above for comments on genomic analysis robustness. Other aspects appear strongly supported experiments to me.

F I don't suggest further extra experiments or data; the authors have done all they can. It is now a subjective editorial matter with respect to the broad significance of the study.

G Improved on version 1, but see comments on B above.

H See B above.

Referee #3 (Remarks to the Author):

The authors' meticulousness in incorporating most of the reviewers' comments and suggestions is commendable. The manuscript's current state is impressive, and I believe the public will also appreciate this piece of work.

I am genuinely excited that the fungus changed genetically while the symbiotic Mycetohabitans did not.

I agree with the authors that the early steps of endosymbiosis should be studied in as many models as possible to gain insights into symbiogenesis across the tree of life.

The only aspect that still needs attention is to make the generated genomic information of the ancestral and evolved fungal-bacterial pairs available to the public. I did not see any explicit reference to this in the actual Ms.

Good job!

Author Rebuttals to First Revision:

Referees' comments:

Referee #1 (Remarks to the Author):

The manuscript has been properly revised. In particular, the addition of genomic data is fine. I have no further comments.

We thank the reviewer for the positive assessment of our work.

Referee #2 (Remarks to the Author):

a) The piece develops a method for transfer of symbionts between isolates of the same species of fungi and have a strong positive selection to maintain the strain to enable evolution. Ultimately the symbiont-host combination becomes a better fit but is still lost when selection is released. Some genomic data underpinning the changes is now presented.

We thank the reviewer for highlighting the establishment of the approach to introduce endosymbionts into fungi and the setup of a positive selection regime leading to adaptation.

b) I still regard the piece as of medium high novelty, because symbiont transfer has quite an old history in insects, and because I was not persuaded by a conceptual advance here (I accept a rather elegant technical advance that enables work in fungi). Whilst there are biological distinctions that certainly make this meritorious science, and it was interesting, I think its impact is to broaden the field of enquiry rather than establish new conceptual knowledge. Of course, how important that is will always be subjective.

I'd also note in opposition to the rebuttal - many insect transinfection studies are not conducted on strongly nutritional symbionts and do not involve host's with housing organs. Most are Wolbachia or Spiroplasma which are facultative, generally not housed in bacteriomes, and have weak nutritional impacts. Of these, Wolbachia sits in a vacuole, but I think Spiroplasma sits in the cytosol directly (though can also be extracellular). Of course it nevertheless remains interesting to expand more widely.

Whilst the title is fungal specific, the abstract still makes it appear transinfection/transplantation study is novel to endosymbiosis research: 'However, the emergence of novel endosymbioses is rare and challenging to uncover in retrospect.' is true of fungi - it is not true of endosymbiosis in general.

The last line of the conclusion. 'The production of rhizoxin by strain NH demonstrates that metabolic capabilities can be transferred to new organisms by artificially inducing an endosymbiosis through implantation and subsequent artificial selection.' similarly is not so novel - Wolbachia from *Drosophila melanogaster*, placed into mosquito cell lines and selected, placed into mosquitoes and released to control dengue in the wet tropics is an example of transfer of traits.

We believe that our work carries novelty compared to existing literature on insect endosymbiogenesis due to the manifold differences in the biological systems and the focus on a different branch of the tree of life. We are convinced that we are still far away from an understanding of the emergence of endosymbioses in general. We are also certain that our targeted single cell approach with high spatio-temporal resolution is important for understanding fundamental dynamics of early endosymbiogenesis.

Regarding the two sentences, we believe these to be factual. While the adjective "rare" is not quantitative, we believe it to be appropriate for an abstract to describe a process which has occurred many times throughout evolutionary history, yet still remains an extremely unlikely event considering the myriad of possible interactions between potential hosts and endosymbionts. Most initial

encounters will fail for many reasons widely discussed in the literature (and cited in the manuscript). This is not only true for fungi but is more fundamental. However, to clarify the phrase, we have changed it in the manuscript. It now reads: “Despite the many examples of known endosymbioses across the tree of life, their de novo emergence is rare and challenging to uncover in retrospect.” We also placed references into the abstract to support the statement and have rewritten the corresponding section in the introduction to be more precise.

We find that the last sentence “The production of rhizoxin by strain NH demonstrates that metabolic capabilities can be transferred to new organisms by artificially inducing an endosymbiosis through implantation and subsequent artificial selection.” is correct, but we deleted the second “artificial”.

c) Data and methodology. The injection and the positive selection through cell sorting are of course novel and technically very elegant.

It was good to see the authors conducting the suggested genomic analysis, but I was not so persuaded of the results from these. It is clear that the bacterial genome is not responsive during this process. It is argued the fungal genome is responsive. However, the timeline of mutations going to high frequency and phenotype changes for fitness don't match well. For instance, in the most intensely monitored - line 4 - in Figure S1, the fitness index for this line increases between generation 4 and 8, but mutations only appear at decent rate from generation 8 (Figure S4). Line 7 appears to achieve improved fitness without any high frequency mutations in the host occurring. The mutations themselves are not obvious strong candidates either - many intergenic or intronic; not impossible to get a phenotype but not clear. These observations leads me to believe that most of what is going on is likely selection for transgenerational plastic response.

We indeed show that the fungal genome changed during the passaging experiment, in contrast to the bacterial one. Figure 5a shows the change in fitness for line 4 throughout all passaging rounds and is compared to the change in the frequency of mutations. Figure S4a, mentioned by the reviewer, shows only the timepoints where the mutations exceed the 50% cut-off. Figure 5a clearly shows that the fitness index of the population correlates with the prevalence of the four mutations, with the highest prevalence and highest fitness in rounds 9 and 10. The lower mutation frequency between generations 4 and 8 is congruent with moderate fitness in the earlier passages. The genomic changes observed through the evolution experiment strongly suggest that the identified genomic changes conferred a phenotypically relevant fitness benefit which led to this remarkable enrichment in the population. We have adapted the manuscript by moving Figure S4c to the Supplementary data to prevent confusing the readers.

As we have openly stated in the manuscript, it is not possible to draw firm conclusions on how much of the adaptation is due to genomic changes and how much is due to epigenetic changes, nor can we pinpoint the underlying molecular mechanisms. It is also possible that some of the early fitness benefits were conferred by various unidentified mutations which did not end up being enriched past the 50% threshold. Analogously, the observation of increased fitness in the absence of highly enriched mutations in line 7 does not exclude the possibility for underlying genomic changes. Co-occurring mutations with similarly strong effect sizes can mask each other because no single mutation becomes enriched past the 50% threshold, even though they cumulatively confer a measurable fitness benefit to the population. For example, we found the mutation in contig 71 at loci 276528/276529 only due to its occurrence in F_{Evo2} (Suppl. Fig. 4 b). When the mutation was tracked over time, it was shown to have been most abundant between round 6 and 8 (Supplementary Data, FEvo Loci). During this time, the mutation may have boosted the population's fitness, but was then outcompeted by the other four mutations before surmounting the 50% threshold.

D,E. See above for comments on genomic analysis robustness. Other aspects appear strongly supported experiments to me.

We thank the reviewer for the positive assessment of these aspects of our work.

F I don't suggest further extra experiments or data; the authors have done all they can. It is now a subjective editorial matter with respect to the broad significance of the study.

We thank for the comment and agree.

G Improved on version 1, but see comments on B above.

H See B above.

Referee #3 (Remarks to the Author):

The authors' meticulousness in incorporating most of the reviewers' comments and suggestions is commendable. The manuscript's current state is impressive, and I believe the public will also appreciate this piece of work.

I am genuinely excited that the fungus changed genetically while the symbiotic Mycetohabitans did not.

I agree with the authors that the early steps of endosymbiosis should be studied in as many models as possible to gain insights into symbiogenesis across the tree of life.

The only aspect that still needs attention is to make the generated genomic information of the ancestral and evolved fungal-bacterial pairs available to the public. I did not see any explicit reference to this in the actual Ms.

Good job!

We thank the reviewer for the positive assessment of our work and the shared excitement. The genomic information is now included in the data availability statement.

Reviewer Reports on the Second Revision:

Referees' comments:

Referee #2 (Remarks to the Author):

This is a second revision of this manuscript.

My main feeling - that the piece is a bit specialist in the overall context of symbiosis research - remains (it is certainly an advance for fungal symbiosis; I don't see the overall conceptual advance or overall technical advance, as the technical advance made is cool but is not required outside fungi).

With respect to the genomics, I guess I am not particularly convinced, though I'll take the point that the strength of conclusion made by the authors is not excessive. I have two final queries on re-reading the manuscript.

i) If one cannot rule out fungal epigenetic effects, one can also not rule out bacterial ones (and I would note that the evolved bacteria introduction to ancestor fungus outperform the original implying they probably exist).

ii) If I read the methods correctly, the genomics changes sought were SNPs vs the reference. Have the authors checked for larger insertion/deletion events in the evolved vs unevolved bacteria? It is quite common in microbial evolution that whole section of genome go (or sometimes arrive) and these changes are functional. For Wolbachia, for instance, virulence and octomom copy number are correlated (work of Luis Teixeira); for others there can be deletions, or rearrangements. I don't think Snippy picks up that type of thing (it looks for changes at a fairly micro-level).

If the authors have the patience, it would be worth checking illumina readdepth coverage of the evolved bacteria against the genome to see if there are gene loss or gain areas (I think rearrangements won't be findable with short reads). This would make their conclusion of no evolutionary change in the bacterium more robust.

Author Rebuttals to Second Revision:

Referees' comments:

Referee #2 (Remarks to the Author):

This is a second revision of this manuscript.

My main feeling - that the piece is a bit specialist in the overall context of symbiosis research - remains (it is certainly an advance for fungal symbiosis; I don't see the overall conceptual advance or overall technical advance, as the technical advance made is cool but is not required outside fungi).

With respect to the genomics, I guess I am not particularly convinced, though I'll take the point that the strength of conclusion made by the authors is not excessive. I have two final queries on re-reading the manuscript.

i) If one cannot rule out fungal epigenetic effects, one can also not rule out bacterial ones (and I would note that the evolved bacteria introduction to ancestor fungus outperform the original implying they probably exist).

We have adapted the manuscript to clarify this point. Please refer to the screenshot of the revised section with the changes indicated.

325 The underexplored genome of this basal fungus makes it difficult to draw firm conclusions about
which molecular mechanisms were critical to adapt to the induced endosymbiosis, and additional
epigenetic effects on either host or endosymbiont side cannot be ruled out. Furthermore, genetic
changes based on rearrangements or larger deletions/duplications may be missed by short read
sequencing. However, multiple genes affected by mutations may be involved in the regulation of
transcription and translation, and in endocytosis (Extended Data Table 4). While exact mechanisms
remain opaque, we could show phenotypically that the system evolved to increase the number of
330 spores with the bacterium, and we identified reduced bacterial load in the spores as an important factor
for the increased germination success (Fig. 4b-d).

Author Deleted: cannot be ruled out.
Author Deleted: Supplementary
Author Deleted: . Mutations

ii) If I read the methods correctly, the genomics changes sought were SNPs vs the reference. Have the authors checked for larger insertion/deletion events in the evolved vs unevolved bacteria? It is quite common in microbial evolution that whole section of genome go (or sometimes arrive) and these changes are functional. For Wolbachia, for instance, virulence and octomom copy number are correlated (work of Luis Teixeira); for others there can be deletions, or rearrangements. I don't think Snippy picks up that type of thing (it looks for changes at a fairly micro-level).

If the authors have the patience, it would be worth checking illumina readdepth coverage of the evolved bacteria against the genome to see if there are gene loss or gain areas (I think rearrangements won't be findable with short reads). This would make their conclusion of no evolutionary change in the bacterium more robust.

We have not found clear evidence for genome rearrangements in the bacteria or the fungus. There are quite many repetitive and transposable elements in the fungal genome which, paired with the lower coverage and incomplete assembly, make it difficult to detect if some of these have changed. For the bacterial genomes, where such changes would be more easily detected even with short read sequencing due to the higher coverage, we did not identify any rearrangements/deletions/duplications despite checking the read coverage. We found differences compared to the published genome, including a larger deletion, but did not detect any changes within our evolution experiment.